# GRANOLA: Adaptive Normalization for Graph Neural Networks

**Moshe Eliasof**[*]
University of Cambridge
me532@cam.ac.uk

**Beatrice Bevilacqua**[*]
Purdue University
bbevilac@purdue.edu

**Carola-Bibiane Schönlieb**
University of Cambridge
cbs31@cam.ac.uk

**Haggai Maron**
Technion & NVIDIA Research
hmaron@nvidia.com

## Abstract

Despite the widespread adoption of Graph Neural Networks (GNNs), these models often incorporate off-the-shelf normalization layers like BatchNorm or InstanceNorm, which were not originally designed for GNNs. Consequently, these normalization layers may not effectively capture the unique characteristics of graph-structured data, potentially even weakening the expressive power of the overall architecture. While existing graph-specific normalization layers have been proposed, they often struggle to offer substantial and consistent benefits. In this paper, we propose GRANOLA, a novel graph-adaptive normalization layer. Unlike existing normalization layers, GRANOLA normalizes node features by adapting to the specific characteristics of the graph, particularly by generating expressive representations of its nodes, obtained by leveraging the propagation of Random Node Features (RNF) in the graph. We provide theoretical results that support our design choices as well as an extensive empirical evaluation demonstrating the superior performance of GRANOLA over existing normalization techniques. Furthermore, GRANOLA emerges as the top-performing method among all baselines in the same time complexity class of Message Passing Neural Networks (MPNNs).

## 1 Introduction

Graph Neural Networks (GNNs) have achieved remarkable success in several application domains [35, 61], showcasing their ability to leverage the rich structural information within graph data. Recently, a plethora of different layer designs has been proposed, each tailored to address specific challenges in the context of GNNs, such as limited expressive power [63, 41, 40] and oversmoothing [46]. Notably, analogously to architectures in other domains [30, 18], these GNN layers are often interleaved with normalization layers, as the integration of normalization methods has empirically proven beneficial in optimizing neural networks, facilitating convergence and enhancing generalization [34, 7, 55].

In practice, most existing GNN architectures employ standard normalization techniques, such as BatchNorm [34], LayerNorm [3], or InstanceNorm [58]. However, these widely adopted normalization techniques were not originally designed with graphs and GNNs in mind. Consequently, they may not effectively capture the unique characteristics of graph-structured data, and can also hinder the expressive power of the overall architecture [11]. These observations highlight the need for graph-specific normalization layers.

---

[*]Equal Contribution

38th Conference on Neural Information Processing Systems (NeurIPS 2024).

Recent works have taken initial steps in this direction, mainly targeting oversmoothing [72, 64, 74] or the expressive power of the overall architecture [11, 14]. Despite the promise shown by these methods, a consensus on a single normalization technique best suited for diverse tasks remains elusive, with no single normalization technique proving clearly superior across all benchmarks and scenarios.

**Our approach.** In this paper, we identify adaptivity to the input graph structure as a desirable property for an effective normalization layer in graph learning. Intuitively, this property ensures that the normalization is tailored to the specific input graph, capturing attributes such as the graph size, node degrees, and connectivity. Importantly, we claim and demonstrate that, given the limitations of practical GNNs, achieving full adaptivity requires expressive architectures that can detect and disambiguate graph substructures, thereby better adapting to input graphs.

Guided by this desirable property, which is absent in existing normalization methods, we introduce our proposed approach – GRANOLA (Graph Adaptive Normalization Layer). GRANOLA aims at dynamically adjusting node features at each layer by leveraging learnable characteristics of the node neighborhood structure derived through the utilization of Random Node Features (RNF) [44, 1, 51, 56, 17]. More precisely, GRANOLA samples RNF and uses them in an additional *normalization* GNN to obtain expressive intermediate node representations. The intermediate representations are then used to scale and shift the node representations obtained by the preceding GNN layer.

We present theoretical results that justify the primary design choices behind our method. Specifically, we demonstrate that GRANOLA is fully adaptive to the input graph, which, in other words, means that GRANOLA can predict different normalization values for non-isomorphic nodes. This property arises from the maximal expressive power of the normalization GNN we employ (MPNN augmented with RNF [1, 51]). In addition, we show that our method inherits this expressive power. Lastly, we show that using standard MPNN layers without RNF within GRANOLA cannot result in a fully adaptive method or in any additional expressive power.

Empirically, we show that GRANOLA significantly and consistently outperforms all existing standard as well as GNN-specific normalization schemes on a variety of different graph benchmarks and architectures. Furthermore, GRANOLA proves to be the best-performing method among all baselines that have the same time complexity as the most widely adopted GNNs, namely the family of MPNNs.

**Contributions.** Our contributions are as follows: (1) We provide an overview of different normalization schemes in graph learning, outlining adaptivity as a desirable property of normalization layers that existing methods are missing, (2) We introduce GRANOLA, a novel normalization technique adjusting node features based on learnable characteristics of their neighborhood structure, (3) We present an intuitive theoretical analysis of our method, giving insights into the design choices we have made, and (4) We conduct an extensive empirical study, providing a thorough benchmarking of existing normalization methods and showcasing the consistently superior performance of GRANOLA.

## 2 Normalization layers for GNNs

### 2.1 Basic setup and definitions

Let $G = (\mathbf{A}, \mathbf{X})$ denote graph with $N \in \mathbb{N}$ nodes, where $\mathbf{A} \in \mathbb{R}^{N \times N}$ is the adjacency matrix and $\mathbf{X} \in \mathbb{R}^{N \times C}$ is the node feature matrix, with $C \in \mathbb{N}$ the feature dimension. Consider a batch of $B \in \mathbb{N}$ graphs encoded by the adjacency matrices $\{\mathbf{A}_b\}_{b=0}^{B-1}$, and, for simplicity, assume that all graphs in the batch have the same number of nodes $N$. We consider a model composed of $L$ GNN layers, with $L \in \mathbb{N}$. Each GNN layer is followed by a normalization layer NORM and an activation function $\phi$. At any layer $\ell \in [L]$, the output of the GNN layer for a batch of graphs consists of (intermediate) node representations, which can be gathered in a matrix $\tilde{\mathbf{H}}^{(\ell)} \in \mathbb{R}^{B \times N \times C}$ (since all graphs have the same number of nodes per our assumption[2]). These undergo a normalization and an activation layer, resulting in node representations denoted by $\mathbf{H}^{(\ell)} \in \mathbb{R}^{B \times N \times C}$, which serve as input of the next GNN layer, with $\mathbf{H}^{(0)} := \mathbf{X}$.[3] Throughout this paper, for any three-dimensional tensor, we use subscripts to denote access to a corresponding dimension. For example, we denote the intermediate node representations of graph $b \in [B]$ by $\tilde{\mathbf{H}}_b^{(\ell)} \in \mathbb{R}^{N \times C}$, and by $\tilde{h}_{b,n,c}^{(\ell)}$ the value of feature $c \in [C]$ in node $n \in [N]$ of graph $b \in [B]$.

---

[2]It is always possible to meet this assumption by padding to the max number of nodes.

[3]For simplicity and without loss of generality, we assume that the feature dimension of all GNN layers is $C$.

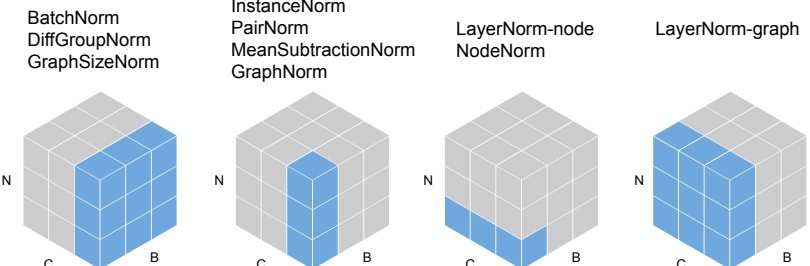

Figure 1: Illustration of normalization layers. We denote by $B$, $N$ and $C$ the number of graphs (batch size), nodes, and channels (node features), respectively. For simplicity of presentation, we use the same number of nodes for all graphs. We color in blue the elements used to compute the statistics employed inside the normalization layer.

Formally, the intermediate, *pre-normalized* node features for the $b$-th graph in the batch are defined as

$$\tilde{\mathbf{H}}_b^{(\ell)} = \text{GNN}_{\text{LAYER}}^{(\ell-1)}\left(\mathbf{A}_b, \mathbf{H}_b^{(\ell-1)}\right). \tag{1}$$

Then, the overall update rule for the batch of $B$ graphs can be written as

$$\mathbf{H}^{(\ell)} = \phi\left(\text{NORM}\left(\tilde{\mathbf{H}}^{(\ell)}; \ell\right)\right), \tag{2}$$

for $\ell \in [L]$. Equation (2) serves as a general blueprint, and in what follows we will show different ways to customize it in order to implement different existing normalization techniques.

We consider normalization layers based on the standardization [38] of their inputs, as this represents the most common choice in widely used normalizations. Generally, a standardization-based normalization layer first shifts each element $\tilde{h}_{b,n,c}^{(\ell)}$ by some mean $\mu_{b,n,c}$, and then scales it by the corresponding standard deviation $\sigma_{b,n,c}$, i.e.,

$$\text{NORM}(\tilde{h}_{b,n,c}^{(\ell)}; \tilde{\mathbf{H}}^{(\ell)}, \ell) = \gamma_c^{(\ell)} \frac{\tilde{h}_{b,n,c}^{(\ell)} - \mu_{b,n,c}}{\sigma_{b,n,c}} + \beta_c^{(\ell)}, \tag{3}$$

where $\gamma_c^{(\ell)}, \beta_c^{(\ell)} \in \mathbb{R}$ are learnable *affine* parameters, that do not depend on $b$ nor $n$.

## 2.2 Current normalization layers for GNNs

The difference among different normalization schemes lies in the set of values used to compute the mean and standard deviation statistics for each element, or, more precisely, across which dimensions of $\tilde{\mathbf{H}}^{(\ell)}$ they are computed. We present them below, and visualize them in Figure 1.

**BatchNorm.** BatchNorm [34] computes the statistics across all nodes and all graphs in the batch, for each feature separately. Therefore, we have

$$\mu_{b,n,c} = \frac{1}{BN}\sum_{b=1}^{B}\sum_{n=1}^{N}\tilde{h}_{b,n,c}^{(\ell)}, \sigma_{b,n,c}^2 = \frac{1}{BN}\sum_{b=1}^{B}\sum_{n=1}^{N}(\tilde{h}_{b,n,c}^{(\ell)} - \mu_{b,n,c})^2, \tag{4}$$

which implies that $\mu_{b,n,c} = \mu_{b',n',c}$ for any $b' \in [B]$ and any $n' \in [N]$ (and similarly for $\sigma_{b,n,c}^2$). Despite the widespread use of BatchNorm in graph learning [63], it is important to recognize scenarios where alternative normalization schemes might be more suitable. A concrete example illustrating this need is presented in Figure 2, focusing on the task of predicting the degree of each node.[4] In this example, BatchNorm, by subtracting the mean computed across the batch, results in negative values for nodes with outputs below the mean. The subsequent ReLU application, as standard practice [63, 37, 41], zeros out these negative values, leading to predictions of 0 for such nodes, irrespective of their actual degree. Additional insights are further discussed in Example C.1 in Appendix C, where it is demonstrated that relying on the affine parameter $\beta_c^{(\ell)}$ to shift the negative output does not provide a definitive solution, since $\beta_c^{(\ell)}$ is the same for all graphs.

---

[4]The node degree is a fundamental feature in graph-based methods, see for example Newman [45].

**InstanceNorm.** InstanceNorm [58] is similar to BatchNorm, but considers each graph separately, that is

$$\mu_{b,n,c} = \frac{1}{N}\sum_{n=1}^{N}\tilde{h}_{b,n,c}^{(\ell)},$$

$$\sigma_{b,n,c}^2 = \frac{1}{N}\sum_{n=1}^{N}(\tilde{h}_{b,n,c}^{(\ell)} - \mu_{b,n,c})^2, \tag{5}$$

which implies that $\mu_{b,n,c} = \mu_{b,n',c}$ for any $n' \in [N]$ (and similarly for $\sigma_{b,n,c}^2$). Notably, the example in Figure 2 can be extended to InstanceNorm by considering all graphs in the batch as disconnected components in a single graph, as we show in Example C.4 in Appendix C.

**LayerNorm.** LayerNorm [3] can be defined in two ways in the context of graphs [24]. The first, which we call *LayerNorm-node*, behaves similarly to LayerNorm in Transformer architectures [59], and computes statistics across the features for each node separately, that is

$$\mu_{b,n,c} = \frac{1}{C}\sum_{c=1}^{C}\tilde{h}_{b,n,c}^{(\ell)},$$

$$\sigma_{b,n,c}^2 = \frac{1}{C}\sum_{c=1}^{C}(\tilde{h}_{b,n,c}^{(\ell)} - \mu_{b,n,c})^2, \tag{6}$$

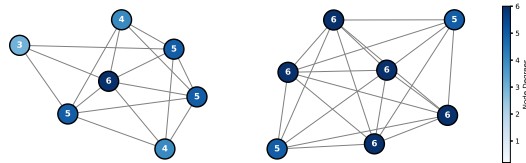

(a) Node degrees, computed by one message-passing layer.

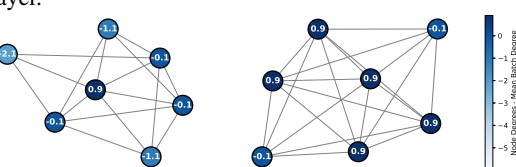

(b) Subtraction of the mean node degree. The features of nodes with degree less than average turn negative.

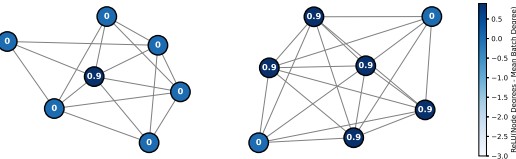

(c) After ReLU, nodes with negative values are mapped to the same value of 0, despite having different degrees.

Figure 2: A batch of two graphs, where subtracting the mean of the node features computed across the batch, as in BatchNorm and related methods, results in the loss of capacity to compute node degrees.

and therefore $\mu_{b,n,c} = \mu_{b,n,c'}$ for any $c' \in [C]$ (and similarly for $\sigma_{b,n,c}^2$). The second variant, which we call *LayerNorm-graph*, is similar to LayerNorm in Computer Vision [62], and computes statistics across the features and across all the nodes in each graph, that is

$$\mu_{b,n,c} = \frac{1}{NC}\sum_{n=1}^{N}\sum_{c=1}^{C}\tilde{h}_{b,n,c}^{(\ell)}, \qquad \sigma_{b,n,c}^2 = \frac{1}{NC}\sum_{n=1}^{N}\sum_{c=1}^{C}(\tilde{h}_{b,n,c}^{(\ell)} - \mu_{b,n,c})^2, \tag{7}$$

and therefore $\mu_{b,n,c} = \mu_{b,n',c'}$ for any $n' \in [N]$ and any $c' \in [C]$ (and similarly for $\sigma_{b,n,c}^2$). Example C.5 in Appendix C presents a motivating example similar to Figure 2 for LayerNorm.

**Graph-Specific normalizations.** Several normalization methods tailored to graphs have been recently proposed. We categorize them based on which dimensions (the batch dimension $B$, the node dimension $N$ within each graph, and the channel dimension $C$) are used to compute the statistics employed within the normalization layer. We illustrate this categorization in Figure 1. DiffGroupNorm [74] and GraphSizeNorm [21] normalize features considering nodes across different graphs, akin to BatchNorm. Similarly to InstanceNorm, PairNorm [72] and MeanSubtractionNorm [64] shift the input by the mean computed per channel across all the nodes in the graph, with differences in the scaling strategies. GraphNorm [11] extends InstanceNorm by incorporating a multiplicative factor to the mean. NodeNorm [75] behaves similarly to LayerNorm-node but only scales the input, without shifting it. We provide additional details in Appendices A and B, and include methods that normalize the adjacency matrix before the GNN layers, which, however, fall outside the scope of this work.

## 3   Method

The following section describes the proposed GRANOLA framework. We start with identifying adaptivity as a desired property in a graph normalization layer.

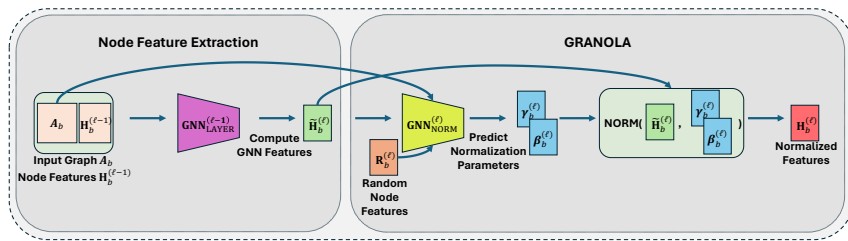

Figure 3: Illustration of a GRANOLA layer. Given node features $\mathbf{H}_b^{(\ell-1)}$ and the adjacency matrix $\mathbf{A}_b$, we feed them to a $\text{GNN}_{\text{LAYER}}^{(\ell-1)}$ to extract intermediate node features $\tilde{\mathbf{H}}_b^{(\ell)}$. Then, we predict normalization parameters using $\text{GNN}_{\text{NORM}}^{(\ell)}$, which takes sampled RNF $\mathbf{R}_b^{(\ell)}$, $\tilde{\mathbf{H}}_b^{(\ell)}$, $\mathbf{A}_b$. Including $\mathbf{R}_b^{(\ell)}$ with $\mathbf{A}_b$ and $\tilde{\mathbf{H}}_b^{(\ell)}$ enhances the expressiveness of GRANOLA ensuring full adaptivity.

**Motivation.** In the previous section, we observed that current normalization schemes used within GNNs are mostly borrowed or adapted from other domains and that they possess two main limitations: (i) Using BatchNorm and InstanceNorm (as well as many methods derived from them, including graph-specific methods) can limit the expressive power of GNNs that use them, preventing them from being able to represent even very simple functions such as computing node degrees; (ii) As shown in previous works, as well as in our experiments in Section 5, these methods do not provide a consistent benefit in downstream performance on different tasks and datasets.

One possible reason for these limitations is that the discussed methods use the same affine parameters in the normalization process, irrespective of the input graph. Crucially, unlike other more structured data types, such as images and time series, graphs do not differ merely in the values associated with each element or time step. Instead, graphs fundamentally vary in the actual connectivity between the nodes. Therefore, it might make sense to employ different (adaptive) normalization schemes based on the features and the connectivity of the input graph. Next, we explore this concept and show that carefully accounting for the graph structure in the normalization process may enhance the expressive power of the GNN, overcoming the previously mentioned failure cases, and providing consistent experimental behavior and overall improved performance.

## 3.1 Design considerations and overview

In an adaptive normalization, instead of using the same affine parameters $\gamma_c^{(\ell)}$ and $\beta_c^{(\ell)}$ (Equation (3)) for all the nodes in all the graphs, the normalization method utilizes specific parameters conditioned on the input graph. Importantly, in other domains, adaptivity in normalization techniques has proven to be a valuable property [19, 33, 48, 76, 47, 49]. In GRANOLA, we achieve this property by generating affine parameters using an additional *normalization* GNN that takes the graph as input, similarly to Hypernetworks [29] that generate weights for another network. Importantly, for full graph adaptivity, where the network can assign different normalization parameters for every pair of non-isomorphic nodes, the normalization GNN should have maximal expressive power. Due to the limited expressive power of MPNNs [41, 63], when designing GRANOLA, we advocate for using a normalization GNN with high expressiveness. Notably, most expressive GNNs come at the expense of significantly increased computational complexity [16, 71, 4, 69, 40, 43]. For this reason, we parameterize the normalization GNN using an MPNN augmented with Random Node Features (RNF), which provide strong function approximation guarantees [1, 51] while retaining the efficiency of standard MPNNs.

## 3.2 GRANOLA

We are now ready to describe our GRANOLA approach, which is based on the utilization of an additional GNN to compute the affine parameters and the integration of RNF [56]. These affine parameters are then used to normalize node representations obtained by the GNN layer preceding the normalization. An overview of GRANOLA is visualized in Figure 3 and described in Algorithm 1.

**GRANOLA layer.** At any given layer $\ell \in [L]$, for each graph $b \in [B]$, we sample RNF $\mathbf{R}_b^{(\ell)} \in \mathbb{R}^{N \times K}$, $K \in \mathbb{N}$, from some joint probability distribution, e.g., Normal, $\mathbf{R}_b^{(\ell)} \sim \mathcal{D}$. Then, we

---

**Algorithm 1** GRANOLA Layer

---

**Input:** Node features $\tilde{\mathbf{H}}_b^{(\ell)} \in \mathbb{R}^{N \times C}$, obtained from $\text{GNN}_{\text{LAYER}}^{(\ell-1)}$.
**Output:** Normalized features per node $n \in [N]$ and channel $c \in [C]$.

1: Sample random node features $\mathbf{R}_b^{(\ell)} \in \mathbb{R}^{N \times K}$.
2: Compute $\mathbf{Z}_b^{(\ell)} = \text{GNN}_{\text{NORM}}^{(\ell)}(\mathbf{A}_b, \tilde{\mathbf{H}}_b^{(\ell)} \oplus \mathbf{R}_b^{(\ell)})$.
3: Compute affine parameters:
   $\gamma_{b,n}^{(\ell)} = f_1^{(\ell)}(z_{b,n}^{(\ell)}), \qquad \beta_{b,n}^{(\ell)} = f_2^{(\ell)}(z_{b,n}^{(\ell)})$.
4: Compute mean and standard-deviation $\mu_{b,n,c}$ and $\sigma_{b,n,c}$ of $\tilde{\mathbf{H}}_b^{(\ell)}$.
5: Return $\gamma_{b,n,c}^{(\ell)} \frac{\tilde{h}_{b,n,c}^{(\ell)} - \mu_{b,n,c}}{\sigma_{b,n,c}} + \beta_{b,n,c}^{(\ell)}$.

---

concatenate $\mathbf{R}_b^{(\ell)}$ with the intermediate node features $\tilde{\mathbf{H}}_b^{(\ell)}$ obtained from $\text{GNN}_{\text{LAYER}}^{(\ell-1)}$ (Equation (1)), and pass the resulting feature matrix through an additional GNN, i.e.,

$$\mathbf{Z}_b^{(\ell)} = \text{GNN}_{\text{NORM}}^{(\ell)}(\mathbf{A}_b, \tilde{\mathbf{H}}_b^{(\ell)} \oplus \mathbf{R}_b^{(\ell)}), \tag{8}$$

where $\oplus$ denotes feature-wise concatenation, and $\text{GNN}_{\text{NORM}}^{(\ell)}$ is a shallow GNN architecture, with $L_{\text{NORM}}^{(\ell)}$ layers. The affine parameters are then computed from $\mathbf{Z}_b^{(\ell)} \in \mathbb{R}^{N \times C}$, that is

$$\gamma_{b,n}^{(\ell)} = f_1^{(\ell)}(z_{b,n}^{(\ell)}), \qquad \beta_{b,n}^{(\ell)} = f_2^{(\ell)}(z_{b,n}^{(\ell)}). \tag{9}$$

where $f_1^{(\ell)}, f_2^{(\ell)}$ are learnable functions (e.g., MLPs), and $\gamma_{b,n}^{(\ell)}, \beta_{b,n}^{(\ell)} \in \mathbb{R}^C$. Then, a GRANOLA layer is defined as

$$\text{NORM}(\tilde{h}_{b,n,c}^{(\ell)}; \tilde{\mathbf{H}}^{(\ell)}, \ell) = \gamma_{b,n,c}^{(\ell)} \frac{\tilde{h}_{b,n,c}^{(\ell)} - \mu_{b,n,c}}{\sigma_{b,n,c}} + \beta_{b,n,c}^{(\ell)}, \tag{10}$$

where $\mu_{b,n,c}$ and $\sigma_{b,n,c}$ are the mean and std of $\tilde{\mathbf{H}}^{(\ell)}$, computed per node across the feature dimension, exactly as in LayerNorm-node (Equation (6)). We highlight here a noticeable difference compared to standard normalization formulations, as presented in Equation (3): in GRANOLA, $\gamma_{b,n,c}^{(\ell)}, \beta_{b,n,c}^{(\ell)}$ not only depend on $c$, but they also have a dependency on $b$ and $n$ and indeed vary for different graphs and nodes. Notably, this is possible because our normalization is adaptive to the input graph. Methods that disregard this information are compelled to use the same learnable normalization values for all graphs, as they operate without knowledge of the specific input graph being considered.

**GRANOLA-NO-RNF.** We consider a variant of GRANOLA that does not sample RNF and instead uses only $\tilde{\mathbf{H}}_b^{(\ell)}$ to obtain $\mathbf{Z}_b^{(\ell)}$ in Equation (8), which therefore becomes

$$\mathbf{Z}_b^{(\ell)} = \text{GNN}_{\text{NORM}}^{(\ell)}(\mathbf{A}, \tilde{\mathbf{H}}_b^{(\ell)}). \tag{11}$$

We refer to this variant as GRANOLA-NO-RNF, as it allows us to directly quantify the contribution of the expressiveness offered by augmenting it with RNF as in Equation (8).

**Complexity.** We conclude by remarking that both GRANOLA and its variant, GRANOLA-NO-RNF, do not impact the asymptotic time and space complexity of standard MPNNs, which remains linear in the number of nodes and edges, as we show in Appendix G.

## 4 Theoretical Analysis

In this section, we explain the main design choices taken in the development of GRANOLA. Specifically, we elaborate on the advantages obtained by utilizing RNF as part of the normalization scheme as opposed to relying solely on the node features. We assume all GNN layers, including those within GRANOLA (Equation (8)) are message-passing layers, as the ones in Xu et al. [63], Morris et al. [41].

**The advantages of using RNF-augmented MPNNs for normalization.** We start by observing that the integration of our GRANOLA in an MPNN allows to easily default to an MPNN augmented

with RNF [56], as we formalize in Proposition E.2 in Appendix E. The idea of the proof lies in the ability of the first normalization layer to default to outputting its input RNF, enabling the rest of the architecture to function as an MPNN augmented with these RNF. The significance of this result lies in the fact that MPNN + GRANOLA inherits the $(\epsilon, \delta)$-universal approximation properties previously proved for MPNNs augmented with RNF [1, 51]. This naturally solves the limitations of existing normalizations identified in Section 2.2, as an MPNN + GRANOLA can leverage universality to approximate such functions. Importantly, the universality of MPNNs augmented with RNF further implies that GRANOLA is *fully adaptive*, as it can approximate any equivariant function on the input graph [1], and therefore can approximate functions returning different normalization values for non-isomorphic nodes.

**Why RNF are necessary?** While GRANOLA employs RNF to compute the normalization affine parameters $\gamma_{b,n,c}^{(\ell)}$ and $\beta_{b,n,c}^{(\ell)}$, the same procedure can be applied without the use of RNF, a variant we denoted as GRANOLA-NO-RNF in Section 3 (Equation (11)). However, we theoretically demonstrate next that an MPNN + GRANOLA-NO-RNF not only loses the universal approximation properties, but is also not more expressive than standard MPNNs.

**Proposition 4.1** (RNF are necessary in GRANOLA for increased expressive power). *Assume our input domain consists of graphs of a specific size. For every MPNN with GRANOLA-NO-RNF (Equation (11)) there exists a standard MPNN with the same expressive power.*

Proposition 4.1 is proven by showing that an MPNN with GRANOLA-NO-RNF can be implemented by a standard MPNN, and, therefore, its expressive power is bounded by the expressive power of a standard MPNN. The proof can be found in Appendix E. This limitation underscores the significance of RNF within GRANOLA. Furthermore, our experiments in Section 5 show that omitting the RNF within the normalization results in degraded performance, as GRANOLA always outperforms GRANOLA-NO-RNF. Finally, we remark that this theoretical result emphasizes the necessity of RNF for increased expressiveness and, consequently, for ensuring full adaptivity to the input graph. That is, any normalization generated by GRANOLA-NO-RNF will be limited by the expressive power of standard MPNNs while GRANOLA does not have this limitation.

**Relation to expressive GNNs.** The results in this section highlight the connection between GRANOLA and expressive GNNs, as our method inherits enhanced expressiveness of MPNNs augmented with RNF. Notably, while MPNNs with RNF have demonstrated theoretical improvements in expressiveness, their practical performance on real-world datasets has not consistently reflected these benefits [22]. Our experimental results indicate that GRANOLA serves as a valuable bridge between theory and practice. Specifically, our findings address the gap between the theoretical expressiveness of MPNNs that use RNF, and their limited practical utility. This is achieved by effectively incorporating RNF within the normalization process rather than simply treating it as an additional input to the MPNN. Importantly, GRANOLA gives rise to a method that is efficient, provably expressive, and performs well in practice, something that, to our knowledge, has never been accomplished by any previous architecture. This provides an additional perspective to our contribution: beyond designing an effective and efficient normalization layer, which is the main scope of our work, we offer a practical approach to realizing the theoretical benefits of RNF. Finally, we conclude this section by remarking that $\text{GNN}_{\text{NORM}}^{(\ell)}$ in Equation (8) can be modified to be any other expressive architecture, and our design choice was mainly guided by the computational practicality of RNF, that allows GRANOLA to offer increased expressive power while retaining the linear complexity of MPNNs, as discussed in Appendix G. We refer the reader to Morris et al. [43] for a survey on expressive methods.

## 5 Experimental Results

In this section, we evaluate the performance of GRANOLA. In particular, we seek to address the following questions: (1) *How does* GRANOLA *compare to other normalization methods?* (2) *Does* GRANOLA *achieve better performance than its natural baselines that also leverage RNF, that is, how important is the graph-adaptivity within* GRANOLA*?* (3) *How does* GRANOLA *compare to its variant* GRANOLA-NO-RNF*, which does not leverage RNF within the normalization, that is, how important are the RNF within* GRANOLA*?* In what follows, we present our main results and refer to Appendices F to H for details and additional experiments, including timings and ablation studies. Our code is available at https://github.com/MosheEliasof/GRANOLA.

Table 2: A comparison to natural baselines, standard and graph normalization layers, demonstrating the practical advantages of GRANOLA. The top three methods are marked by **First**, **Second**, **Third**.

| Method ↓ / Dataset → | MOLESOL RMSE ↓ | MOLTOX21 ROC-AUC ↑ | MOLBACE ROC-AUC ↑ | MOLHIV ROC-AUC ↑ |
|---|---|---|---|---|
| **NATURAL BASELINES** | | | | |
| GIN + BatchNorm + RNF-PE [56] | 1.052±0.041 | 75.14±0.67 | 74.28±3.80 | 75.98±1.63 |
| GIN + RNF-NORM | 1.039±0.040 | 75.12±0.92 | 77.96±4.36 | 77.61±1.64 |
| **STANDARD NORMALIZATION LAYERS** | | | | |
| GIN + BatchNorm [63] | 1.173±0.057 | 74.91±0.51 | 72.97±4.00 | 75.58±1.40 |
| GIN + InstanceNorm [58] | 1.099±0.038 | 73.82±0.96 | 74.86±3.37 | 76.88±1.93 |
| GIN + LayerNorm-node [3] | 1.058±0.024 | 74.81±0.44 | 77.12±2.70 | 75.24±1.71 |
| GIN + LayerNorm-graph [3] | 1.061±0.043 | 75.03±1.24 | 76.49±4.07 | 76.13±1.84 |
| GIN + Identity | 1.164±0.059 | 73.34±1.08 | 72.55±2.98 | 71.89±1.32 |
| **GRAPH NORMALIZATION LAYERS** | | | | |
| GIN + PairNorm [72] | 1.084±0.031 | 73.27±1.05 | 75.11±4.24 | 76.18±1.47 |
| GIN + MeanSubtractionNorm [64] | 1.062±0.045 | 74.98±0.62 | 76.36±4.47 | 76.37±1.40 |
| GIN + DiffGroupNorm [74] | 1.087±0.063 | 74.48±0.76 | 75.96±3.79 | 74.37±1.68 |
| GIN + NodeNorm [75] | 1.068±0.029 | 73.27±0.83 | 75.67±4.03 | 75.50±1.32 |
| GIN + GraphNorm [11] | 1.044±0.027 | 73.54±0.80 | 73.23±3.88 | 78.08±1.16 |
| GIN + GraphSizeNorm [21] | 1.121±0.051 | 74.07±0.30 | 76.18±3.52 | 75.44±1.51 |
| GIN + SuperNorm [12] | 1.037±0.044 | 75.08±0.98 | 75.12±3.38 | 76.55±1.76 |
| GIN + GRANOLA-NO-RNF | 1.088±0.032 | 75.87±0.72 | 76.23±2.06 | 77.09±1.49 |
| GIN + GRANOLA | 0.960±0.020 | 77.19±0.85 | 79.92±2.56 | 78.98±1.17 |

**Baselines.** For each task, we consider the following baselines: (1) Standard normalization layers, (2) Graph-designated normalization layers and (3) Our natural baselines, namely: (i) GRANOLA-NO-RNF, which corresponds to the variant of GRANOLA that uses only the intermediate features, without RNF, inside the normalization (Equation (11)) to assess the practical importance of RNF for full graph-adaptivity; (ii) RNF as a positional encoding (RNF-PE), where we augment only the initial input features with RNF, as in Sato et al. [56]; (iii) The normalization that uses only RNF, without any message passing layer inside the normalization, to compute the normalization affine parameters $\gamma_{b,n,c}^{(\ell)}$ and $\beta_{b,n,c}^{(\ell)}$. This is achieved by considering $\mathbf{Z}_b^{(\ell)} = \mathbf{R}_b^{(\ell)}$ in Equation (8). We denote this baseline by RNF-NORM and remark that it corresponds to a version of GRANOLA without graph-adaptivity. For a fair comparison, in all the baselines, as well as in our method, we employ GIN [63] or GINE [32] layers, which are maximally-expressive within the MPNN family. Furthermore, we compare GRANOLA to GNNs in the same complexity class and refer the reader to Appendix H.8 for additional comparisons.

**ZINC.** We experiment with the ZINC-12K molecular dataset [57, 28, 21], where the goal is to regress the solubility of molecules. As can be seen from Table 1, GRANOLA achieves significantly lower (better) mean-absolute-error compared to existing standard and graph-designated normalization layers, while outperforming all its natural baselines. Moreover, GRANOLA emerges as the best-performing model with the same time complexity of a standard MPNN.

**OGB.** We test our GRANOLA on the OGB collection [31], and report its performance in Table 2. Notably, GRANOLA consistently improves over existing standard normalization methods while retaining similar asymptotic complexity. For instance, on MOLBACE, we achieve an accuracy of 79.92% compared to 72.97% when using BatchNorm, an improvement of 6.95%. Compared with graph-designated normalization methods, GRANOLA also offers a consistent improvement. As an example, on MOLESOL we obtain a RMSE of (lower is bet-

Table 1: Comparison of GRANOLA with various baselines on the ZINC-12K dataset. All methods obey to the 500k parameter budget. The top three methods are marked by **First**, **Second**, **Third**.

| Method | ZINC (MAE ↓) |
|---|---|
| **NATURAL BASELINES** | |
| GIN + BatchNorm + RNF-PE [56] | 0.1621±0.014 |
| GIN + RNF-NORM | 0.1562±0.013 |
| **STANDARD NORMALIZATION LAYERS** | |
| GIN + BatchNorm [63] | 0.1630±0.004 |
| GIN + InstanceNorm [58] | 0.2984±0.017 |
| GIN + LayerNorm-node [3] | 0.1649±0.009 |
| GIN + LayerNorm-graph [3] | 0.1609±0.014 |
| GIN + Identity | 0.2209±0.018 |
| **GRAPH NORMALIZATION LAYERS** | |
| GIN + PairNorm [72] | 0.3519±0.008 |
| GIN + MeanSubtractionNorm [64] | 0.1632±0.021 |
| GIN + DiffGroupNorm [74] | 0.2705±0.024 |
| GIN + NodeNorm [75] | 0.2119±0.017 |
| GIN + GraphNorm [11] | 0.3104±0.012 |
| GIN + GraphSizeNorm [21] | 0.1931±0.016 |
| GIN + SuperNorm [12] | 0.1574±0.018 |
| GIN + GRANOLA-NO-RNF | 0.1497±0.008 |
| GIN + GRANOLA | 0.1203±0.006 |

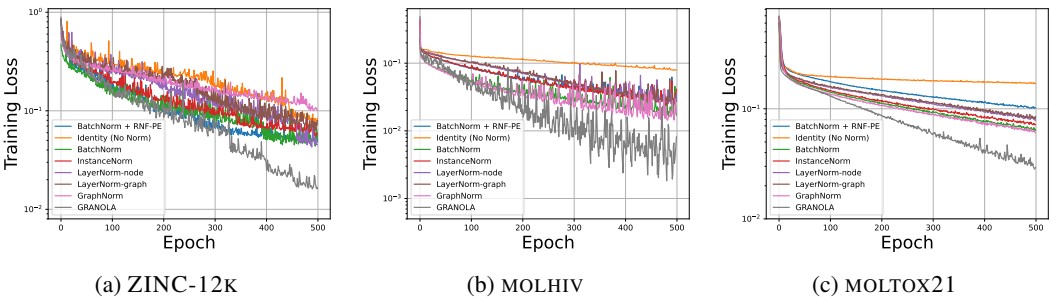

|                | (a) ZINC-12K | (b) MOLHIV | (c) MOLTOX21 |

Figure 4: Training convergence of GRANOLA compared with existing normalization techniques show that GRANOLA achieves faster convergence and overall lower (better) MAE.

Table 3: Empirical results of GSN with GRANOLA show that GRANOLA can also improve the performance of expressive methods.

| Method | ZINC
MAE ↓ | MOLESOL
RMSE ↓ | MOLTOX21
ROC-AUC ↑ | MOLBACE
ROC-AUC ↑ | MOLHIV
ROC-AUC ↑ |
|---|---|---|---|---|---|
| GSN [10] | 0.1010±0.010 | 1.003±0.037 | 76.08±0.79 | 77.40±2.92 | 80.39±0.90 |
| GSN + GRANOLA | 0.0766±0.008 | 0.941±0.024 | 77.84±0.63 | 80.41±2.07 | 81.12±0.79 |

ter) 0.960, compared to the second best graph normalization layer, GraphNorm, that achieves 1.044. It is also noteworthy to mention the performance gap between GRANOLA and its natural baselines, such as RNF-PE, which emphasizes the practical benefit of graph-adaptivity within GRANOLA.

**TUDatasets.** We experimented with popular datasets from the TUD [42] repository. Our results are reported in Table 14 in Appendix H.8, suggesting that GRANOLA consistently achieves higher accuracy compared to its natural baselines, as well as standard and graph-designated normalization techniques. For example, on the NCI109 dataset, GRANOLA achieves an accuracy of 83.7%, compared to the second-best normalization technique, GraphNorm, with an accuracy 82.4%.

**Training convergence of GRANOLA.**   In addition to improved downstream task performance being one of the main benefits of a normalization layer, accelerated training convergence is also an important desired property [34, 11]. Figure 4 shows that GRANOLA offers faster convergence and lower MAE compared to other methods.

**Combining GRANOLA with expressive methods.** Since our goal is to understand the impact of normalization layers, our experiments focus on studying the benefit of augmenting standard and well-understood MPNNs with GRANOLA. However, it is also interesting to understand if expressive, domain-expert approaches such as GSN [10] can also benefit from GRANOLA. To this end, we augment GSN with GRANOLA, and report the results in Table 3. These results further underscore the versatility of GRANOLA, which can be coupled with any GNN layer and improve its performance.

**Discussion.** Our experimental results cover standard normalization layers, as well as graph normalization methods, evaluated on 11 datasets from diverse sources, and applied to various tasks. Throughout all experiments, a common theme is the performance *consistency* of GRANOLA. Specifically, GRANOLA always improves over its natural baselines and other normalization techniques across all datasets. In contrast, other existing methods exhibit less clear trends in their performance. While some methods achieve competitive results on certain datasets, they may struggle on others. Notable examples are GraphNorm and PairNorm, which, despite offering improved performance compared to BatchNorm on most of the OGB datasets, show worse results on ZINC-12K. Furthermore, standard normalization layers also lack consistency. As an example, consider InstanceNorm, which is beneficial in some OGB datasets, yet, it does not offer favorable results on ZINC-12K.

## 6   Conclusions

In this paper, we discuss the existing landscape of feature normalization techniques in Graph Neural Networks (GNNs). Despite recent advances in designing graph normalization techniques, the optimal choice remains unclear, with methods not offering consistent performance improvements across tasks. To address this challenge, we identify a desirable property of graph normalization layers, namely

adaptivity to the input graph, and argue that it can be obtained only with expressive architectures. To incorporate this property, we present GRANOLA, a normalization layer that adjusts node features based on the specific input graph, leveraging Random Node Features (RNF). Our theoretical analyses support the design choices of GRANOLA, demonstrating its increased expressiveness. Empirical evaluations across diverse benchmarks consistently highlight the superior performance of GRANOLA over existing normalization methods, as well as other baselines with the same time complexity.

**Limitations and impact.** Although GRANOLA exhibits promising results, there are areas for potential improvement in future research. For instance, investigating alternative designs for $\text{GNN}_{\text{NORM}}^{(\ell)}$ could further enhance the performance, as, in certain datasets, there is still a gap between GRANOLA and expressive GNNs. Additionally, exploring ways to reduce memory and time complexity (which are still linear in the number of nodes) represents an important avenue for future research. Furthermore, by improving the performance of GNN through GRANOLA we envision a positive impact in domains such as drug discovery.

## Acknowledgements

ME is funded by the Blavatnik-Cambridge fellowship, the Cambridge Accelerate Programme for Scientific Discovery, and the Maths4DL EPSRC Programme. HM is the Robert J. Shillman Fellow and is supported by the Israel Science Foundation through a personal grant (ISF 264/23) and an equipment grant (ISF 532/23).

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

# A  Additional Related Work

**Standard normalization layers.** BatchNorm [34] is arguably one of the most widely used normalization schemes in deep learning. Despite its success, there is little consensus on the exact reason behind the improvements it generally yields. While some studies argue that the effectiveness of BatchNorm lies in its ability to control the problem of internal covariate shift [34, 2], other works [7, 55] attribute the success to the promotion of faster convergence. Similarly to the other domains, also in graph-learning, BatchNorm normalizes each channel (feature) independently by computing the mean and standard deviation across all the elements in the batch, i.e., across all the nodes in all the graphs in the batch. Another normalization approach that was adopted in graph-learning is InstanceNorm [58], which was originally introduced in image style-transfer tasks to remove image-specific contrast and style information. In the context of graphs, InstanceNorm normalizes each channel (feature) independently by computing the mean and standard deviation across all nodes within each graph separately. Additionally, LayerNorm [3] was originally proposed for recurrent models, and it is also widely used in Transformers architectures [59]. In the context of graph learning, LayerNorm can take two different versions: Layernorm-node and LayerNorm-graph [23]. The former normalizes each node independently, by computing the mean and standard deviation across all channels within each node separately, while the latter uses the mean and standard deviation across all the channels of all nodes in the entire graph. We visualize these variants in Figure 1. The common theme of these three discussed methods is that they were not originally designed with graph-learning in mind. Next, we discuss graph-designated normalization techniques.

**Graph-designated normalization layers.** PairNorm [72] was introduced to mitigate the issue of oversmoothing in GNNs, where repeated graph convolutions eventually make node representations indistinguishable from one and the other. The key idea of PairNorm is to ensure that the total pairwise feature distances remain constant across layers, preventing the features of distant nodes from becoming overly similar or indistinguishable. Yang et al. [64] proposed a different understanding of the oversmoothing issue in GNNs: the authors show that upon initialization GNNs suffer from oversmoothing, but GNNs learn to anti-oversmooth during training. Based on this conclusion, the authors design MeanSubtractionNorm, which removes the mean from the inputs, without rescaling the features. When coupled with GCN [37], MeanSubtractionNorm leads to a revised Power Iteration that converges to the Fiedler vector, resulting in a coarse graph partition and faster training. To enhance the performance of GNNs on large graphs, Li et al. [39] propose MessageNorm, a method that normalizes the aggregated message for each node before combining it with its node features, demonstrating experimental benefits. DiffGroupNorm [74] was designed to alleviate oversmoothing by softly clustering nodes and normalizing nodes within the same group to increase their smoothness while also separating node distributions among different groups. Zhou et al. [75] explored the impact of transformation operators, such as normalizations, that transform node features after the propagation step in GNNs, and argue that they tend to amplify node-wise feature variance. This effect is shown to lead to performance drop in deep GNNs. To mitigate this issue, they introduced NodeNorm, which scales the feature vector of each node by a root of the standard deviation of its features. This approach also shares similarities with LayerNorm-node. While the aforementioned normalization techniques have made significant strides in mitigating the oversmoothing phenomenon in GNNs, other normalization methods have been proposed to address problems beyond this specific concern. Cai et al. [11] showed that InstanceNorm serves as a preconditioner for GNNs, thus accelerating their training convergence. However, it also provably degrades the expressiveness of the GNNs for regular graphs. To address this issue, they propose GraphNorm, which builds on top of InstanceNorm by adding a learnable multiplicative factor to the mean for each channel. GraphSizeNorm [21] was proposed based on promising empirical results, and aims at normalizing the node features by dividing them by the graph size, before applying a standard BatchNorm layer. UnityNorm [13] was introduced to learn graph normalization by optimizing a weighted combination of existing techniques, including LayerNorm-node, InstanceNorm, and BatchNorm. Finally, SuperNorm [12] first computes subgraph-specific factors, encompassing the number of nodes and the eigenvalues of the neighborhood-induced subgraph for each node. These subgraph-specific factors are then explicitly embedded at the beginning and end of a standard BatchNorm layer. This approach ensures that any arbitrary GNN layer becomes at least as powerful as the 1-WL test, while simultaneously mitigating the oversmoothing issue. We conclude this paragraph by acknowledging the existence of normalization techniques that focus on normalizing the adjacency matrix before the GNN layers. One common example is the symmetric normalization which is used in GCN [37] and subsequent

works. Despite their wide use, especially on node-level tasks, these normalizations may fail to capture structural information and lead to less expressive architectures. For instance, a common practice of normalizing the adjacency matrix by dividing its rows by the sum of their entries is equivalent to employing a mean aggregator over the neighbors, which has been shown to fail to distinguish certain non-isomorphic nodes [63]. For this reason, in this work, we consider only normalization layers applied to node features after every GNN layer.

**AdaIN and StyleGANs.** Adaptive instance normalization (AdaIN) [33, 19, 26] was originally proposed for real-time arbitrary style transfer. Given a content input and a style input, AdaIN adjusts the mean and variance of the content input to match those of the style input. This allows for the transfer of stylistic features from the style input to the content input in a flexible and efficient manner. Motivated by style-transfer literature, Karras et al. [36] introduced StyleGANs as a variant of GANs [27] which differs in the generator architecture. More precisely, the generator in StyleGANs starts from a learned constant input and adjusts the "style" of the image at each convolution layer based on the latent code through the usage of AdaIN. Similarly to StyleGANs, we adjust node features at every layer through the usage of noise, i.e., random node features, akin to the role played by the latent code in StyleGANs.

**Random Node Features (RNF) in GNNs.** Random input features were used in the context of GNNs to improve the expressive power of Message Passing Neural Networks (MPNNs) [44, 51, 1, 56, 17]. Importantly, MPNNs augmented with random node features have been shown to be universal (with high probability) in Puny et al. [51], Abboud et al. [1], a result we naturally inherit when GRANOLA defaults to simply utilizing random node features. Notably, despite the universality results, GNNs augmented with RNF do not consistently improve performance on real-world datasets [22]. We show instead that GRANOLA consistently leads to improved performances.

## B  Comparison of Different Normalizations

In this section we present the different normalization techniques that have been specifically proposed for graph data and highlight their connection to standard normalization schemes (i.e., BatchNorm, InstanceNorm, LayerNorm-node, LayerNorm-graph) whenever this is present.

**PairNorm.** PairNorm [72] first centers node features by subtracting the mean computed per channel across all the nodes in the graph, similarly to InstanceNorm. Then, it scales the centered vector by dividing it by the square root of the mean (computed across nodes) of the norm of each node feature vector, where the norm is computed over the channel dimension. That is, the center operation has equation:

$$\tilde{h}_{b,n,c}^{(\ell),\text{center}} = \tilde{h}_{b,n,c}^{(\ell)} - \mu_{b,n,c},$$

where $\mu_{b,n,c}$ is as in InstanceNorm (Equation (5)), while the scale operation can be written as

$$\text{NORM}(\tilde{h}_{b,n,c}^{(\ell)}; \tilde{\mathbf{H}}^{(\ell)}, \ell) = s \cdot \frac{\tilde{h}_{b,n,c}^{(\ell),\text{center}}}{\sqrt{\frac{1}{N} \sum_{n=1}^{N} \|\tilde{h}_{b,n}^{(\ell),\text{center}}\|_2^2}}, \tag{12}$$

with $s \in \mathbb{R}$ a hyperparameter and the norm is

$$\|\tilde{h}_{b,n}^{(\ell),\text{center}}\|_2^2 = \sum_{c=1}^{C} |\tilde{h}_{b,n,c}^{(\ell),\text{center}}|^2.$$

**MeanSubtractionNorm.** MeanSubtractionNorm [64] is similar to InstanceNorm, but it simply shifts its input without dividing it by the standard deviation (and without affine parameters). That is

$$\text{NORM}(\tilde{h}_{b,n,c}^{(\ell)}; \tilde{\mathbf{H}}^{(\ell)}, \ell) = \tilde{h}_{b,n,c}^{(\ell)} - \mu_{b,n,c} \tag{13}$$

and $\mu_{b,n,c}$ as Equation (5).

**MessageNorm.** MessageNorm [39] couples normalization with the GNN layer. In particular, it defines the update rule of $\mathbf{H}^{(\ell)}$ as follows,

$$h_{b,n}^{(\ell)} = \phi \left( \text{MLP} \left( h_{b,n}^{(\ell-1)} + s\|h_{b,n}^{(\ell-1)}\|_2 \frac{\|m_{b,n}^{(\ell-1)}\|_2}{\|m_{b,n}^{(\ell-1)}\|_2} \right) \right) \tag{14}$$

where $s \in \mathbb{R}$ a hyperparameter and $m_{b,n}^{(\ell-1)}$ is the message of node $n \in [N]$ in graph $b \in [B]$, which can be defined as

$$m_{b,v,u}^{(\ell)} = \rho^{(\ell)}(h_{b,v}^{(\ell)}, h_{b,u}^{(\ell)})$$
$$m_{b,u}^{(\ell)} = \zeta^{(\ell)}(\{m_{b,u,v}^{(\ell)} | u \in \mathcal{N}_v^b\})$$

with $\rho^{(\ell)}, \zeta^{(\ell)}$ learnable functions such as MLPs and $\mathcal{N}_v^b$ neighbors of node $v$ in graph $b$.

**DiffGroupNorm.** DiffGroupNorm [74] first softly clusters nodes and the normalizes nodes within the same cluster by means of BatchNorm. That is, for each graph $b \in [B]$ in the batch, it computes a soft cluster assignment as

$$\mathbf{S}_b^{(\ell)} = \text{softmax}(\tilde{\mathbf{H}}_b^{(\ell)}\mathbf{W}^{(\ell)})$$

where $\mathbf{W}^{(\ell)} \in \mathbb{R}^{C \times D}$ with $D \in \mathbb{N}$ the number of clusters, and therefore $\mathbf{S}_b^{(\ell)} \in \mathbb{R}^{N \times D}$. Let us denote by $\mathbf{S}^{(\ell)} \in \mathbb{R}^{B \times N \times D}$ the matrix containing the cluster assignments for all graphs in the batch, where the $b$-th row of $\mathbf{S}^{(\ell)}$ is $\mathbf{S}_b^{(\ell)}$. DiffGroupNorm computes a linear combination of the output of BatchNorm applied to each cluster (where the feature is weighted by the assignment value), and adds the result to the input embedding, that is

$$\text{NORM}(\tilde{h}_{b,n,c}^{(\ell)}; \tilde{\mathbf{H}}^{(\ell)}, \ell) = \tilde{h}_{b,n,c}^{(\ell)} + \lambda \sum_{i=1}^{D} \text{BATCHNORM}(s_{b,n,i}^{(\ell)}\tilde{h}_{b,n,c}^{(\ell)}; \tilde{\mathbf{H}}^{(\ell)}, \ell), \qquad (15)$$

where BATCHNORM leverages Equation (4), and $\lambda \in \mathbb{R}$ is a hyperparameter. Notably, the term $\tilde{h}_{b,n,c}^{(\ell)}$ in Equation (15) is similar to a skip connection, with the difference lying in the fact that skip connections usually add the output of the previous layer $\ell - 1$ after the norm instead of the output of the current layer before the norm.

**NodeNorm.** NodeNorm [75] is similar to LayerNorm-node, but it simply divides its input by a root of its standard deviation, without shifting it and without affine parameters:

$$\text{NORM}(\tilde{h}_{b,n,c}^{(\ell)}; \tilde{\mathbf{H}}^{(\ell)}, \ell) = \frac{\tilde{h}_{b,n,c}^{(\ell)}}{\sigma_{b,n,c}^{\frac{1}{p}}} \qquad (16)$$

with $\sigma$ as in Equation (6) and $p \in \mathbb{N}$ a hyperparameter.

**GraphNorm.** GraphNorm [11] builds upon InstanceNorm by adding an additional learnable parameter $\alpha^{(\ell)} \in \mathbb{R}^C$ which is the same for all nodes and graphs. The equation can be written as

$$\text{NORM}(\tilde{h}_{b,n,c}^{(\ell)}; \tilde{\mathbf{H}}^{(\ell)}, \ell) = \gamma_c^{(\ell)} \frac{\tilde{h}_{b,n,c}^{(\ell)} - \alpha_c^{(\ell)}\mu_{b,n,c}}{\sigma_{b,n,c}} + \beta_c^{(\ell)}, \qquad (17)$$

and $\mu_{b,n,c}, \sigma_{b,n,c}$ as Equation (5).

**GraphSizeNorm.** GraphSizeNorm [21] normalizes the node features by dividing them by the graph size, before applying a standard BatchNorm:

$$\text{NORM}(\tilde{h}_{b,n,c}^{(\ell)}; \tilde{\mathbf{H}}^{(\ell)}, \ell) = \frac{\tilde{h}_{b,n,c}^{(\ell)}}{\sqrt{N}}. \qquad (18)$$

**UnityNorm.** UnityNorm [13] consists of a weighted combination of four normalization techniques, where the weights $\lambda_1, \lambda_2, \lambda_3, \lambda_4 \in \mathbb{R}$ are learnable. That is

$$\begin{aligned}
\text{NORM}(\tilde{h}_{b,n,c}^{(\ell)}; \tilde{\mathbf{H}}^{(\ell)}, \ell) =& \lambda_1\text{LAYERNORM-NODE}(\tilde{h}_{b,n,c}^{(\ell)}; \tilde{\mathbf{H}}^{(\ell)}, \ell) + \\
& \lambda_2\text{ADJACENCYNORM}(\tilde{h}_{b,n,c}^{(\ell)}; \tilde{\mathbf{H}}^{(\ell)}, \ell) + \\
& \lambda_3\text{INSTANCENORM}(\tilde{h}_{b,n,c}^{(\ell)}; \tilde{\mathbf{H}}^{(\ell)}, \ell) + \\
& \lambda_4\text{BATCHNORM}(\tilde{h}_{b,n,c}^{(\ell)}; \tilde{\mathbf{H}}^{(\ell)}, \ell)
\end{aligned} \qquad (19)$$

where LAYERNORM-NODE, INSTANCENORM and BATCHNORM leverage respectively Equations (4) to (6). ADJACENCYNORM computes the statistics for each node across all features of all its neighbors, that is

$$\mu_{b,n,c} = \frac{1}{|\mathcal{N}_n^b|C} \sum_{u \in \mathcal{N}_n^b} \sum_{c=1}^{C} \tilde{h}_{b,u,c}^{(\ell)}, \qquad \sigma_{b,n,c}^2 = \frac{1}{|\mathcal{N}_n^b|C} \sum_{u \in \mathcal{N}_n^b} \sum_{c=1}^{C} (\tilde{h}_{b,u,c}^{(\ell)} - \mu_{b,n,c})^2.$$

**SuperNorm.** SuperNorm [12] embeds subgraph-specific factors into BatchNorm. First, for each node $n$ in graph $b$ it extracts the subgraph induced by its neighbors, denoted as $S_{b,n}$. Then, it computes the so called subgraph-specific factors for each subgraph, which are the output of an hash function over the number of nodes in the subgraph and its eigenvalues. That is the subgraph-specific factor of $S_{b,n}$ is

$$\xi(S_{b,n}) = \text{Hash}(\phi(|V_{S_{b,n}}|), \psi(\text{Eig}_{S_{b,n}}))$$

where $|V_{S_{b,n}}|$ denotes the number of nodes in $S_{b,n}$ and $\text{Eig}_{S_{b,n}}$ its eigenvalues, with $\phi$ and $\psi$ injective functions.

Subgraph-specific factors for all subgraphs in a graph $b$ are collected into a vector $M_b^G \in \mathbb{R}^{N \times 1}$

$$M_b^G = [\xi(S_{b,1}); \xi(S_{b,2}); \ldots; \xi(S_{b,N})]$$

which is used to obtain two additional vectors $M_b^{SN}, M_b^{RC} \in \mathbb{R}^{N \times 1}$ defined as

$$M_b^{SN} = \left[ \frac{\xi(S_{b,1})}{\sum_{n=1}^N \xi(S_{b,n})}; \frac{\xi(S_{b,2})}{\sum_{n=1}^N \xi(S_{b,n})}; \ldots; \frac{\xi(S_{b,N})}{\sum_{n=1}^N \xi(S_{b,n})} \right]$$
$$M_b^{RC} = M_b^G \odot M_b^{SN}.$$

where $\odot$ denotes the element-wise product. Then, the normalization computes the segment average of $\tilde{\mathbf{H}}_b^{(\ell)}$ for each graph $b$, denoted as $\tilde{\mathbf{H}}_b^{(\ell),\text{segment}} \in \mathbb{R}^{N \times C}$, where a row $n$ is defined as

$$\tilde{h}_{b,n}^{(\ell),\text{segment}} = \sum_{n=1}^{N} \tilde{h}_{b,n}^{(\ell)}.$$

Then, the input $\tilde{\mathbf{H}}_b^{(\ell)}$ is calibrated by injecting the subgraph-specific factors as well as the graph statistics obtained via $\tilde{\mathbf{H}}_b^{(\ell),\text{segment}}$. That is,

$$\tilde{\mathbf{H}}_b^{(\ell),\text{calibration}} = \tilde{\mathbf{H}}_b^{(\ell)} + \mathbf{W}^{(\ell)} \tilde{\mathbf{H}}_b^{(\ell),\text{segment}} \odot (M_b^{RC} \mathbf{1}_C^\top)$$

where $\mathbf{1}_C \in \{1\}^{C \times 1}$ and $\mathbf{W}^{(\ell)} = \mathbf{1}_N \mathbf{w}^{(\ell)\top}$ with $\mathbf{1}_N \in \{1\}^{N \times 1}$ and $\mathbf{w}^{(\ell)} \in \mathbb{R}^{C \times 1}$ is a learnable parameter. After injecting subgraph-specific factors, the normalization layer performs BatchNorm on the calibration features, that is

$$\tilde{h}_{b,n,c}^{(\ell),\text{CS}} = \text{BATCHNORM}(\tilde{h}_{b,n,c}^{(\ell),\text{calibration}}; \tilde{\mathbf{H}}^{(\ell),\text{calibration}})$$

Finally, subgraph-specific factors are embedded after BatchNorm, by computing

$$\mathbf{H}_b^{(\ell)} = \phi\left( \tilde{\mathbf{H}}_b^{(\ell),\text{CS}} \odot (\mathbf{1}_N \gamma^{(\ell)\top} + \mathbb{P}_b^{(\ell)})/2 + \mathbf{1}_N \beta^{(\ell)\top} \right) \tag{20}$$

where $\gamma^{(\ell)}, \beta^{(\ell)} \in \mathbb{R}^{C \times 1}$ are learnable affine parameters, and $\mathbb{P}_b^{(\ell)} \in \mathbb{R}^{N \times C}$ is a matrix where each entry is obtained as

$$\mathbb{P}_{b,n,c}^{(\ell)} = (M_{b,n,c}^{RE})^{\mathbf{w}_{RE,c}^{(\ell)}}$$

where $\mathbf{w}_{RE}^{(\ell)} \in \mathbb{R}^C$ is a learnable parameter and $M_b^{RE} \in \mathbb{R}^{N \times C}$ is obtained as

$$M_b^{RE} = \frac{M_b^{RC}}{\sum_{n=1}^N M_{b,n}^{RC}} \mathbf{1}_C^\top.$$

Equation (20) represents the output of SuperNorm, followed by an activation function $\phi$.

# C   Additional Motivating Examples

In the following we elaborate on the failure cases of existing normalization layers. Throughout this section, we will assume that all GNN layers are message-passing layers, and, in particular, we focus on the maximally expressive MPNN layers presented in Morris et al. [41], which have been shown to be as expressive as the 1-WL test. In particular Equation (1) can be rewritten as follows,

$$\tilde{\mathbf{H}}_b^{(\ell)} = \mathbf{H}_b^{(\ell-1)}\mathbf{W}_1^{(\ell-1)} + \mathbf{A}_b\mathbf{H}_b^{(\ell-1)}\mathbf{W}_2^{(\ell-1)}. \tag{21}$$

*Example* C.1 (BatchNorm reduces GNN capabilities to compute node degrees (Figure 2)). Consider the following task: Given a graph $G$, for each node predict its degree. Assume that our batch contains $B$ graphs, and, for simplicity, assume that they all have the same number of nodes $N$.[5] Assuming such graphs do not have initial node features, we follow standard practice [63] and initialize $\mathbf{H}_b^{(0)}$ for each graph $b \in [B]$ as a vector of ones, i.e., $\mathbf{H}_b^{(0)} = \mathbf{1} \in \mathbb{R}^{N \times 1}$. The output of the first GNN layer is already the degree of each node, or a function thereof:

$$\mathbf{H}_b^{(1)} = \phi\left(\text{NORM}\left(\mathbf{H}_b^{(0)}\mathbf{W}_1^{(0)} + \mathbf{A}_b\mathbf{H}_b^{(0)}\mathbf{W}_2^{(0)}; \ell\right)\right), \tag{22}$$

where $\mathbf{W}_1^{(0)} \in \mathbb{R}^{1 \times C}$, $\mathbf{W}_2^{(0)} \in \mathbb{R}^{1 \times C}$, are learnable weight matrices, and $C \in \mathbb{N}$ is the hidden feature dimensionality. Note that since the input is one dimensional, all output channels behave identically. We consider the case where $C = 1$. Importantly, in our example, we have that $\mathbf{H}_b^{(0)}\mathbf{W}_1^{(0)}$ is the same for all nodes in all graphs in the batch, because $\mathbf{H}_b^{(0)}$ and $\mathbf{W}_1^{(0)}$ are the same for all nodes and graphs. Therefore, the term $\mathbf{H}_b^{(0)}\mathbf{W}_1^{(0)}$ acts as a bias term in Equation (22). Thus, for each node, the output of the first GNN layer is simply a linear function of the node degree, which is computed by the term $\mathbf{A}_b\mathbf{H}_b^{(0)}$ in Equation (22). Now, consider the normalization layer NORM applied to the output of this function, and assume for now that there are no affine parameters. First, the BatchNorm normalization layer subtracts the mean computed across all nodes in all graphs, as shown in Equation (4). For all nodes having an output smaller than the mean, this subtraction returns a negative number, which is zeroed out by the application of the ReLU which follows the normalization. Therefore, assuming no further layers, for these nodes the prediction can only be 0 regardless of the actual degree, and therefore is incorrect. In Figure 2, we provide a concrete example where this limitation occurs.

*Remark* C.2 (Deeper networks are also limited). It is important to note that even when the number of layers is greater than one, and assuming $C = 1$ in all layers, the problem persists, because the analysis can be applied to every subsequent layer's output by simply noticing that subtracting the mean will always zero out features less than the mean.

*Remark* C.3 (BN with Affine transformation is also limited). We highlight that the inclusion of learnable affine parameters in BatchNorm, while potentially mitigating the problem during training by means of a large affine parameter $\beta_c^{(1)}$ that shifts the negative outputs before the ReLU, does not offer a definitive solution. Indeed, it is always possible to construct a test scenario where one graph has nodes with degrees significantly smaller than those seen while training, for which the learned shift $\beta_c^{(1)}$ is not sufficiently large to make the output of the normalization layer positive.

It is important to mention that the aforementioned example potentially serves as a failure case for other methods that are based on removing the mean $\mu_{b,n,c}$ as calculated by BatchNorm in Equation (4). For instance, GraphSizeNorm [21] also suffers from the same issue, as this normalization technique scales the node features by the number of nodes before applying BatchNorm. Furthermore, Example C.1 can be adjusted to induce a failure case for the normalization operation (excluding the skip link) in DiffGroupNorm [74], which is also based on BatchNorm. This can be achieved by ensuring that a subset of nodes has a degree less than the average within all the clusters to which it is (softly) assigned.

Example C.1 can be easily adapted to represent a failure case for InstanceNorm. The adaptation involves treating the entire batch as a single graph, which can be further refined to avoid having disconnected components by adding a small number of edges connecting them.

---

[5]This assumption is included only to simplify the notation, but can be removed without affecting the results.

*Example* C.4 (InstanceNorm reduces GNN capabilities to compute node degree (Figure 2 considering all graphs as disconnected components in a single graph)). Consider the following task: Given a graph $G$, for each node predict its degree. Assuming the graph does not have initial node features, we follow standard practice [63] and initialize $\mathbf{H}_b^{(0)} = \mathbf{1} \in \mathbb{R}^{N \times 1}$. The output of the first GNN layer is already the degree of each node, or a function thereof. Now, consider the normalization layer NORM applied to the output of this function. First, the InstanceNorm normalization layer subtracts the mean computed across all nodes within each graph in the batch, as shown in Equation (5). For all nodes having an output smaller than the mean within their graph, this subtraction returns a negative number, which is zeroed out by the application of the ReLU activation function, applied following the normalization. Therefore, assuming no further layers, for these nodes the prediction can only be 0 regardless of the actual degree, and therefore is incorrect. Similarly to BatchNorm, the inclusion of learnable affine parameters as well as the stacking of additional single-channel layers is not sufficient to solve the problem.

The aforementioned limitation extends directly to other graph-designed normalizations based on InstanceNorm, such as PairNorm and GraphNorm[6] as they also shifts the input in the same way, based on the mean $\mu_{b,n,c}$ as defined in Equation (5).

*Example* C.5 (LayerNorm-node reduces GNN capabilities to compute node degree (Figure 5)). Consider the following task: Given a graph $G$, for each node predict its degree. Consider a graph $G$ consisting of a path of 3 nodes. Assuming the graph does not have initial node features, we initialize $\mathbf{H}_b^{(0)} = \mathbf{1} \in \mathbb{R}^{N \times 1}$, with $b = 0$ as we only have one graph. We will assume that $\mathbf{W}_1^{(0)} = \mathbf{0}$ (Equation (21)). Therefore, the output of the first GNN layer can be written as:

$$\mathbf{H}_b^{(1)} = \phi\left(\text{NORM}\left(\mathbf{A}_b \mathbf{H}_b^{(0)} \mathbf{W}_2^{(0)}; \ell\right)\right), \quad (23)$$

which, for each node, is a vector having at each entry the degree of the node multiplied by a learnable weight entry. Compare now one of the node having degree $d \in \mathbb{N}$, denoted as $v$ with the node having degree $2d$, denoted as $u$. Clearly, we have that the vector of node features in $u$ is equal to two times the vector of node features in $v$. Therefore, by subtracting the mean computed across the channel dimension, as per Equation (6), we obtain

$$h_{b,u,c}^{(1)} = h_{b,v,c}^{(1)}$$

for all $c \in [C]$. This implies that it is not possible to then distinguish the degree of these two nodes.

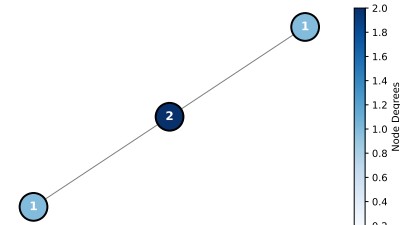

(a) Node degrees, computed by one message-passing layer.

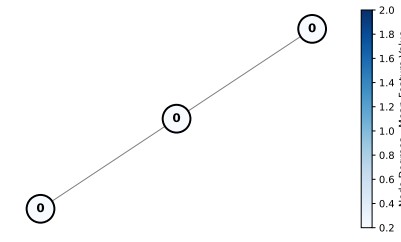

(b) After removing the mean computed on the feature dimension, all nodes are mapped to the same value of 0, despite having different degrees.

Figure 5: A graph, where subtracting the mean of the node features computed on the feature dimension, as in LayerNorm-node and related methods, results in the loss of capacity to compute node degrees.

*Remark* C.6 (Deeper networks are also limited). It is important to note that even when the number of layers is greater than one, assuming $\mathbf{W}_1^{(\ell-1)} = \mathbf{0}$ in all GNN layers (Equation (21)), the problem persists. Indeed, in the next layer, both nodes $v$ and $u$ will aggregate neighbors having identical features. Therefore the vector of node features in $u$ will still be equal to two times the vector of node features in $v$, a difference that is removed by the subtraction of the mean computed per node.

## D    Variants of GRANOLA

In the main paper, we have additionally considered GRANOLA-NO-RNF, a variant of GRANOLA that does not sample RNF and instead uses only $\tilde{\mathbf{H}}_b^{(\ell)}$ (Equation (11)). In the following, we present an additional variant.

---

[6]Assuming the learned shift parameter in GraphNorm is not all zeros.

**GRANOLA-MS.** Equation (10) can be specialized to the case where the affine parameters correspond to the mean and standard deviation computed per node across the feature dimension of $\mathbf{Z}_b^{(\ell)}$. That is,

$$\gamma_{b,n,c}^{(\ell)} = \frac{1}{C}\sum_{c=1}^{C} z_{b,n,c}^{(\ell)}, \qquad \beta_{b,n,c}^{(\ell)} = \frac{1}{C}\sum_{c=1}^{C}(z_{b,n,c}^{(\ell)} - \gamma_{b,n,c}^{(\ell)})^2. \tag{24}$$

We refer to this variant of Equation (10) which uses the pre-defined mean and standard deviation functions in Equation (24) for $f_1^{(\ell)}, f_2^{(\ell)}$ as GRANOLA-MS, where MS stands indeed for mean and standard deviation. The idea behind GRANOLA-MS is to explicitly align the node-wise mean and variance of $\tilde{\mathbf{H}}_b^{(\ell)}$ to match those of $\mathbf{Z}_b^{(\ell)}$. Note that, while GRANOLA-MS follows a predefined realization of $f_1$ and $f_2$ by using the mean and standard deviation, it is still a learnable normalization technique, as $\mathbf{Z}_b^{(\ell)}$ is learned in Equation (8). In Appendix H.8, we provide empirical comparisons with this specialized case.

# E  Proofs

**Theorem E.1** (Existing normalization techniques limit MPNNs' expressivity). *Let $f$ be a stacking of GIN layers with non-linear activations followed by sum pooling. Let $f^{norm}$ be the architecture obtained by adding InstanceNorm or BatchNorm without affine parameters. Then $f^{norm}$ is strictly less expressive than $f$.*

*Proof.* All non-isomorphic graphs that can be distinguished by $f^{norm}$ can clearly be distinguished by $f$ as the normalization only shifts and scale its input. To show that $f^{norm}$ is strictly less expressive than $f$, consider two CSL graphs with different numbers of nodes. These are distinguishable by $f$. However, applying InstanceNorm to the output of GIN results in a zero matrix, as shown in Cai et al. [11, Proposition 4.1]. Similarly, if the batch consists of these two graphs, applying BatchNorm results in a zero matrix. Since the output of the normalization is a zero matrix, they are indistinguishable by $f^{norm}$, concluding our proof. □

**Proposition E.2** (MPNN with GRANOLA can implement MPNN with RNF). *Let $f^{norm}$ be a stacking of MPNN layers interleaved with GRANOLA normalization layers (Equation (10)), followed by activation functions. There exists a choice of hyperparameters and weights such that $f^{norm}$ defaults to MPNN + RNF [1].*

*Proof.* We only need to show a choice of hyperparameters and weights that makes an MPNN + GRANOLA default to an MPNN + RNF, which is the model obtained by performing message passing on the input graph where the initial node features are concatenated to RNF.

Since RNF in an MPNN + GRANOLA are introduced in the GRANOLA layer, we choose the first MPNN layer, which precedes any normalization (Equation (1)), to simply repeat its inputs using additional channels. In the case of GraphConv layers, this is easily obtained by $\mathbf{W}_1^{(0)} = (\mathbf{I}\ \mathbf{I})$ and $\mathbf{W}_2^{(0)} = \mathbf{0}$, with $\mathbf{I} \in \{0,1\}^{C\times C}$. With these choices, $\tilde{\mathbf{H}}_b^{(1)}$ in Equation (1) becomes $\tilde{\mathbf{H}}_b^{(1)} = \mathbf{H}_b^{(0)} \oplus \mathbf{H}_b^{(0)} = \mathbf{X}_b \oplus \mathbf{X}_b$. Notably, this concatenation is only introduced to make the dimension of $\tilde{\mathbf{H}}_b^{(1)}$ match the dimension of the concatenation of the initial node features with RNF having the same dimension.

Consider the first GRANOLA layer, $\ell = 1$. It is sufficient to set the activation functions inside $\mathrm{GNN}_{\mathrm{NORM}}^{(1)}$ to be the identity function and to properly set the weights of $\mathrm{GNN}_{\mathrm{NORM}}^{(1)}$ (Equation (8)) and $f_1^{(1)}, f_2^{(1)}$ (Equation (9)) such that the normalization layer returns $\mathbf{H}_b^{(0)} \oplus \mathbf{R}_b^{(1)}$, with $\mathbf{R}_b^{(1)} \in \mathbb{R}^{N\times K}$ and $K$ chosen such that $K = C$. For example, if $\mathrm{GNN}_{\mathrm{NORM}}^{(1)}$ is composed by a single GraphConv layer [41] (with an identity activation function), we have

$$\begin{aligned}
\mathbf{Z}_b^{(1)} &= (\tilde{\mathbf{H}}_b^{(1)} \oplus \mathbf{R}_b^{(1)})\mathbf{W}_1^{\mathrm{NORM},(1)} + \mathbf{A}_b(\tilde{\mathbf{H}}_b^{(1)} \oplus \mathbf{R}_b^{(1)})\mathbf{W}_2^{\mathrm{NORM},(1)} \\
&= (\mathbf{H}_b^{(0)} \oplus \mathbf{H}_b^{(0)} \oplus \mathbf{R}_b^{(1)})\mathbf{W}_1^{\mathrm{NORM},(1)} + \mathbf{A}_b(\mathbf{H}_b^{(0)} \oplus \mathbf{H}_b^{(0)} \oplus \mathbf{R}_b^{(1)})\mathbf{W}_2^{\mathrm{NORM},(1)},
\end{aligned}$$

then for $\text{GNN}_{\text{NORM}}^{(1)}$ is sufficient to choose $\mathbf{W}_2^{\text{NORM},(1)} = \mathbf{0}$, $\mathbf{W}_1^{\text{NORM},(1)} = \begin{pmatrix} \mathbf{0} & \mathbf{0} \\ \mathbf{I} & \mathbf{0} \\ \mathbf{0} & \mathbf{I} \end{pmatrix}$, where $\mathbf{I} \in \{0,1\}^{C \times C}$ is the identity matrix. For $f_1^{(1)}, f_2^{(1)}$ its is sufficient to set $f_1^{(1)}$ to always return a zero vector, and $f_2^{(1)}$ to be the identity function. With these choices, Equation (2) becomes

$$\mathbf{H}^{(1)} = \phi\left(\mathbf{H}_b^{(0)} \oplus \mathbf{R}_b^{(1)}\right) = \phi\left(\mathbf{X}_b \oplus \mathbf{R}_b^{(1)}\right),$$

which represents the input of the next GNN layer, and matches the input of an MPNN + RNF. Therefore, we are only left to show that subsequent applications of GRANOLA layers behave as the identity function, since the GNN layers will instead behave as those in the MPNN + RNF.

After the first GRANOLA layer, $\ell > 1$, it is sufficient to set $\text{GNN}_{\text{NORM}}^{(\ell)}$ (Equation (8)) and $f_1^{(\ell)}, f_2^{(\ell)}$ (Equation (9)) to return its input $\tilde{\mathbf{H}}^{(\ell)} \in \mathbb{R}^{N \times C}$ (while discarding $\mathbf{R}_b^{(\ell)} \in \mathbb{R}^{N \times K}$). Assuming a single layer in it (with an identity activation function), this can be accomplished by setting $\mathbf{W}_2^{\text{NORM},(\ell)} = \mathbf{0}$ and $\mathbf{W}_1^{\text{NORM},(\ell)} = \left(\begin{smallmatrix} \mathbf{I} \\ \mathbf{0} \end{smallmatrix}\right)$, $f_1^{(\ell)}$ to always return a zero vector, and $f_2^{(\ell)}$ to be the identity function. With these choices, Equation (10) becomes

$$\text{NORM}(\tilde{h}_{b,n,c}^{(\ell)}; \tilde{\mathbf{H}}^{(\ell)}, \ell) = \tilde{h}_{b,n,c}^{(\ell)}.$$

Therefore, these two steps imply that an MPNN with GRANOLA implements an MPNN with RNF. $\square$

**Corollary E.3** (MPNN + GRANOLA is universal with high probability). *Let $\Omega_N$ be a compact set of graphs with $N \in \mathbb{N}$ nodes and $g$ a continuous permutation-invariant graph function defined over $\Omega_N$. Let $f^{\text{norm}}$ be a stacking of MPNN layers interleaved with GRANOLA normalization layers (Equation (10)) followed by activation functions. Then, if random node features inside GRANOLA are sampled from a continuous bounded distribution with zero mean and finite variance, for all $\epsilon, \delta > 0$, there exist a choice of hyperparameters and weights such that, $\forall G = (\mathbf{A}, \mathbf{X}) \in \Omega$,*

$$P(|g(\mathbf{A}, \mathbf{X}) - f^{\text{norm}}(\mathbf{A}, \mathbf{X})| \leq \epsilon) \geq 1 - \delta \tag{25}$$

*where $f^{\text{norm}}(\mathbf{A}, \mathbf{X})$ is the output for the considered choice of hyperparameters and weights.*

*Proof.* The proof follows by showing the choice of hyperparameters and weights which makes an MPNN augmented with GRANOLA satisfy the assumptions of Puny et al. [51, Proposition 1]. In all our GRANOLA layers, $\mathbf{R}_b^{(\ell)} \in \mathbb{R}_b^{N \times K}$ is drawn from a continuous and bounded distribution for any graph $b$ in a batch of $B$ graphs. Therefore, we only need to show that the overall architecture can default to an MPNN augmented with these RNF. This follows from Proposition E.2. $\square$

**Proposition 4.1** (RNF are necessary in GRANOLA for increased expressive power). *Assume our input domain consists of graphs of a specific size. For every MPNN with GRANOLA-NO-RNF (Equation (11)) there exists a standard MPNN with the same expressive power.*

*Proof.* The proof follows by showing that an MPNN with GRANOLA-NO-RNF can be implemented by a standard MPNN (without any normalization), which therefore represents an upper-bound of its expressive power.

We assume that all the MPNN layers are maximally expressive MPNNs layers of the form of GraphConv [41], and that all activation functions inside GRANOLA-NO-RNF are ReLU activation functions. Since the convolution layers are the same in both $f^{\text{norm}}$ and $f$, we are only left to show that an MPNN (without normalization, with ReLU activations) can implement a layer of GRANOLA-NO-RNF. Recall that a single GRANOLA layer can be written as

$$\text{NORM}(\tilde{h}_{b,n,c}^{(\ell)}; \tilde{\mathbf{H}}^{(\ell)}, \ell) = f_1^{(\ell)}(z_{b,n}^{(\ell)})_c \frac{\tilde{h}_{b,n,c}^{(\ell)} - \mu_{b,n,c}}{\sigma_{b,n,c}} + f_2^{(\ell)}(z_{b,n}^{(\ell)})_c, \tag{26}$$

and, since we are not considering RNF, $\mathbf{Z}_b^{(\ell)}$ is obtained with Equation (11), recalling that $\text{GNN}_{\text{NORM}}^{(\ell)}$ is also composed by MPNN layers interleaved by ReLU activation functions per our assumption. We denote the number of layers of this GNN by $L_{\text{NORM}}^{(\ell)}$. We next show how to obtain Equation (26) using multiple layers of an MPNN which takes as input $\tilde{\mathbf{H}}^{(\ell)}$. We will denote intermediate layers of this MPNN as $\hat{\mathbf{H}}^{(t)}, t \geq 1$.

**First Step, compute $\mathbf{Z}_b^{(\ell)}$.** The first layers of this MPNN are used to obtain $\mathbf{Z}_b^{(\ell)}$ given $\tilde{\mathbf{H}}_b^{(\ell)}$, effectively mimicking $\text{GNN}_{\text{NORM}}^{(\ell)}$. Since we will later need also $\tilde{\mathbf{H}}_b^{(\ell)}$, we will use the last feature dimensions in every layer representation to simply copy it in every subsequent layer. Importantly, however, simply copying $\tilde{\mathbf{H}}_b^{(\ell)}$ in the last dimensions using an identity weight matrix may not be sufficient, as the application of ReLU non-linearity would clip negative entries to 0. Therefore, we copy both $\tilde{\mathbf{H}}_b^{(\ell)}$ and $-\tilde{\mathbf{H}}_b^{(\ell)}$ in the last dimensions, and at the end recover $\tilde{\mathbf{H}}_b^{(\ell)}$ as $\tilde{\mathbf{H}}_b^{(\ell)} = \phi(\tilde{\mathbf{H}}_b^{(\ell)}) - \phi(-\tilde{\mathbf{H}}_b^{(\ell)})$, with $\phi$ the ReLU activation.

For $t = 1$, the MPNN layer is simply responsible of replicating its input $\tilde{\mathbf{H}}_b^{(\ell)}$, so that we can later use the first $C$ channels to obtain $\mathbf{Z}_b^{(\ell)}$ and the last to later recover $\tilde{\mathbf{H}}_b^{(\ell)}$. Therefore,

$$\hat{\mathbf{H}}_b^{(1)} = \phi(\tilde{\mathbf{H}}_b^{(\ell)}\hat{\mathbf{W}}_1^{(1)} + \mathbf{A}_b\tilde{\mathbf{H}}_b^{(\ell)}\hat{\mathbf{W}}_2^{(1)})$$

with $\hat{\mathbf{W}}_1^{(1)} = (\,\mathbf{I}\ \mathbf{I}\ -\mathbf{I}\,)$ and $\hat{\mathbf{W}}_2^{(1)} = \mathbf{0}$, with $\mathbf{I} \in \{0,1\}^{C \times C}$ the identity matrix, and $\phi$ is the identity activation function. This means that $\hat{\mathbf{H}}_b^{(1)} = \tilde{\mathbf{H}}_b^{(\ell)} \oplus \tilde{\mathbf{H}}_b^{(\ell)} \oplus -\tilde{\mathbf{H}}_b^{(\ell)}$. For $t > 1$ and $t \le L_{\text{NORM}}^{(\ell)} + 1$, we need to mimic $\text{GNN}_{\text{NORM}}^{(\ell)}$ from Equation (11) on the first $C$ dimensions. This is achievable by

$$\hat{\mathbf{H}}_b^{(t+1)} = \phi(\hat{\mathbf{H}}_b^{(t)}\hat{\mathbf{W}}_1^{(t)} + \mathbf{A}_b\hat{\mathbf{H}}_b^{(t)}\hat{\mathbf{W}}_2^{(t)})$$

with $\hat{\mathbf{W}}_1^{(t)} = \begin{pmatrix} \mathbf{W}_1^{\text{NORM},(t)} & \mathbf{0} & \mathbf{0} \\ \mathbf{0} & \mathbf{I} & \mathbf{0} \\ \mathbf{0} & \mathbf{0} & \mathbf{I} \end{pmatrix}$ and $\hat{\mathbf{W}}_2^{(t)} = \begin{pmatrix} \mathbf{W}_2^{\text{NORM},(t)} & \mathbf{0} & \mathbf{0} \\ \mathbf{0} & \mathbf{I} & \mathbf{0} \\ \mathbf{0} & \mathbf{0} & \mathbf{I} \end{pmatrix}$, where $\mathbf{W}_1^{\text{NORM},(t)}, \mathbf{W}_2^{\text{NORM},(t)}$ are exactly the same as the corresponding weights of $\text{GNN}_{\text{NORM}}^{(\ell)}$, $\phi$ is the ReLU activation. Therefore, after $L_{\text{NORM}}^{(\ell)} + 1$ layers, we have

$$\hat{\mathbf{H}}_b^{(L_{\text{NORM}}^{(\ell)}+1)} = \mathbf{Z}_b^{(\ell)} \oplus \phi(\tilde{\mathbf{H}}_b^{(\ell)}) \oplus \phi(-\tilde{\mathbf{H}}_b^{(\ell)})$$

We can then use an additional MPNN layer $t$ to recover $\tilde{\mathbf{H}}_b^{(\ell)}$ by setting $\hat{\mathbf{W}}_1^{(t)} = \begin{pmatrix} \mathbf{I} & \mathbf{0} \\ \mathbf{0} & \mathbf{I} \\ \mathbf{0} & -\mathbf{I} \end{pmatrix}$, $\hat{\mathbf{W}}_2^{(1)} = \mathbf{0}$, and obtain

$$\hat{\mathbf{H}}_b^{(L_{\text{NORM}}^{(\ell)}+2)} = \mathbf{Z}_b^{(\ell)} \oplus \tilde{\mathbf{H}}_b^{(\ell)}$$

Finally, we rely on the ability of MLPs to memorize a finite number of input-output pairs (see Yun et al. [68] and Yehudai et al. [65, Lemma B.2]) to implement Equation (26) given its input. This is achieved by making the three subsequent layers of the MPNN behave as a 3-layer MLP (with ReLU activations) by simply zeroing out $\hat{\mathbf{W}}_2^{(t)}$. In this way, we have obtained Equation (26) through MPNN layers only. $\qquad\square$

## F   Experimental Details

We implemented GRANOLA using Pytorch [50] (BSD-style license) and Pytorch-Geometric [24] (MIT license). We ran our experiments on NVIDIA RTX3090 and RTX4090 GPUs, both having 24GB of memory. We performed hyperparameter tuning using the Weight and Biases framework [6]. To sample the random node features, we followed Abboud et al. [1] and used a standard Gaussian with mean 0 and variance 1, sampling random node features for each forward pass. For each node, we sample independently a number of random features equal to the channel dimension of $\tilde{h}_{b,n}^{(\ell)}$, that is $K$ is equal to $C$ in Equation (8). Recall that when augmenting a GNN architecture with GRANOLA, we have two types of networks: (i) the main downstream network, and (ii) the normalization network returning $\mathbf{Z}_b^{(\ell)}$ (see Equation (8)). In both networks, and throughout all datasets that do not contain edge features (e.g., in TUDatasets), we employ GIN layers [63] to perform message passing. In case edge features are available, such as in ZINC, we use the GINE variant of GIN, as prescribed in Dwivedi et al. [21]. For brevity, we refer to both as GIN throughout the paper. Our MLPs are composed of two linear layers with ReLU non-linearities. Our normalization network is kept small and comprises a number of layers tuned in $\{1, 2, 3\}$ and an embedding dimension that is the same as its input $\tilde{\mathbf{H}}^{(\ell)}$. Each experiment is repeated for 5 different seeds, and we report the average and standard deviation result. Details of hyperparameter grid for each dataset can be found in the following subsections.

### F.1 ZINC-12k

We consider the dataset splits proposed in Dwivedi et al. [21], and use the Mean Absolute Error (MAE) both as loss and evaluation metric. For all models, we used a batch size tuned in $\{32, 64, 128\}$. To optimize the model we use the Adam optimizer with initial learning rate of $0.001$, which is decayed by $0.5$ every $300$ epochs. The maximum number of epochs is set to $500$. The test metric is computed at the best validation epoch. The downstream network is composed of a number of layers in $\{4, 6\}$, with an embedding dimension tuned in $\{32, 64\}$.

### F.2 OGB datasets

We consider the scaffold splits proposed in Hu et al. [31], and for each dataset we used the loss and evaluation metric prescribed therein. In all experiments, we used the Adam optimizer with initial learning rate of $0.001$. We tune the batch size in $\{64, 128$. We employ a learning rate scheduler that that follows the procedure prescribed in Bevilacqua et al. [4]. We also consider dropout in between layers with probabilities in $\{0, 0.5\}$. The downstream network has the number of layers in $\{4, 6\}$ with embedding dimensions in $\{32, 64, 128\}$. The maximum number of epochs is set to $500$ for all models. The test metric is computed at the best validation epoch.

### F.3 TUDatasets

For all the experiments with datasets from the TUDatasets repository, we followed the evaluation procedure proposed in Xu et al. [63], consisting of 10-fold cross validation and metric at the best averaged validation accuracy across the folds. The downstream network is composed of a number of layers tuned in $\{4, 6\}$ layers with embedding dimension in $\{32, 64\}$. We use the Adam optimizer with learning rate tuned in $\{0.01, 0.001\}$. We consider batch size in $\{32, 64, 128\}$, and trained for $500$ epochs.

## G Complexity and Runtimes

**Complexity.** As described in Section 3, and specifically in Equation (10), our GRANOLA takes random node feature and hidden learned node features, and propagates them using a GNN backbone denoted by $\text{GNN}_{\text{NORM}}$ to compute intermediate (expressive) features, which are then used to calculate the normalization statistics, as shown in Equation (9). The calculation can be implemented either by considering their mean and standard deviation, as in GRANOLA-MS, or more generally by employing an MLP in GRANOLA, as described in Equation (24) and Equation (9), respectively. In our experiments, $\text{GNN}_{\text{NORM}}^{(\ell)}$ in GRANOLA is also an MPNN, similar to the downstream backbone model. Therefore, including our GRANOLA layers does not change the asymptotic computational complexity of the architecture, which remains within the computational complexity of MPNNs (e.g., Morris et al. [41], Xu et al. [63]). Specifically, each MPNN layer is linear in the number of nodes $|V|$ and edges $|E|$. Since a single GRANOLA layer is composed by $L_{\text{NORM}}^{(\ell)}$ MPNN layers, assuming it the same for all $\ell$, it has a time complexity of $\mathcal{O}(L_{\text{NORM}} \cdot (|V| + |E|))$. Every downstream MPNN layer in our framework uses a GRANOLA normalization layer, and therefore, assuming $L$ downstream MPNN layers, the overall complexity of an MPNN augmented with GRANOLA is $\mathcal{O}((L \cdot L_{\text{NORM}}) \cdot (|V| + |E|))$, compared to the complexity of an MPNN without GRANOLA which amounts to $\mathcal{O}(L \cdot (|V| + |E|))$. In practice, $L_{\text{NORM}}$ is a hyperparameter between 1 to 3 and can therefore be considered a constant.

**Runtimes.** While the asymptotic complexity remains linear with respect to the number of nodes and edges in the graph, as in standard MPNNs, our GRANOLA requires some additional computations due to the hidden layers in the normalization mechanism. To measure the impact of these additional layers, we measure the required training and inference times of GRANOLA, whose results are reported in Table 4. Specifically, we report the average time per batch measured on a Nvidia RTX-2080 GPU. For a fair comparison, in all methods, we use the same number of layers, batch size and number of channels. Our results indicate that while GRANOLA requires additional computational time, it is still a fraction of the cost of more complex methods like Subgraph GNNs ($5.2\times$ faster than efficient expressive models like Subgraph GNNs), while yielding favorable downstream performance. These results indicate that GRANOLA offers a strong tradeoff between performance and cost.

Table 4: Average batch runtimes on a Nvidia RTX-2080 GPU of GRANOLA and other methods, with 8 layers, batch size of 128, and 128 channels on the OGBG-MOLHIV DATASET. For reference, we also include the measured metric, which is ROC-AUC.

| Method | Training Time (ms) | MOLHIV Inference Time (ms) | ROC-AUC ↑ |
|---|---|---|---|
| **MPNN** | | | |
| GIN + BatchNorm [63] | 4.12 | 3.23 | 75.58±1.40 |
| **Subgraph GNNs** | | | |
| DSS-GNN (EGO+) [4] | 69.58 | 49.88 | 76.78±1.66 |
| **Natural Baselines** | | | |
| GIN + BatchNorm + RNF-PE [56] | 5.16 | 4.54 | 75.98±1.63 |
| GIN + RNF-NORM | 7.34 | 5.59 | 77.61±1.64 |
| GIN + GRANOLA-NO-RNF | 10.82 | 9.21 | 77.09±1.49 |
| GIN + GRANOLA-MS | 11.15 | 9.37 | 78.84±1.22 |
| GIN + GRANOLA | 11.24 | 9.55 | 78.98±1.17 |

Table 5: GRANOLA coupled with additional backbones. Incorporating GRANOLA enhances performance across all backbones.

| Method | ZINC MAE ↓ | MOLHIV ROC-AUC ↑ |
|---|---|---|
| GCN | 0.367±0.011 | 76.06±0.97 |
| GCN + GRANOLA | 0.233±0.005 | 77.54±1.10 |
| GAT | 0.384±0.007 | 76.0±0.80 |
| GAT + GRANOLA | 0.254±0.009 | 77.39±1.03 |
| GPS | 0.070±0.004 | 78.80±1.01 |
| GPS + GRANOLA | 0.062±0.006 | 79.21±1.26 |

# H   Additional Experimental Results

## H.1   GRANOLA Coupled with Additional Backbones

In this subsection, we evaluate the performance of GRANOLA when coupled with additional backbones beyond GIN and GSN as presented in Section 5. Specifically, we evaluate its integration with GCN [37], GAT [60], and GPS [53], using GRANOLA as the normalization layer. Table 5 shows that adding GRANOLA results in improved performance regardless of the architecture. These results underscore the versatility of GRANOLA, which can be coupled with any GNN layer and improve its performance.

## H.2   Using other Expressive Mechanisms in GRANOLA

As explained in Sections 3.1 and 4, in this paper, we chose to use RNF within GRANOLA. Incorporating RNF into our GRANOLA allows it to fully adapt to the input graph, providing different affine parameters for non-isomorphic nodes. Full adaptivity is lost when removing RNF and using a standard MPNN as GNN$_{\text{NORM}}$, as in GRANOLA-NO-RNF. This is because GRANOLA-NO-RNF is not more expressive than an MPNN (Proposition 4.1), and thus, there exist non-isomorphic nodes that will get the same representation (and the same affine parameters). However, any other most expressive architecture used as GNN$_{\text{NORM}}$ would achieve the same full adaptivity, and our choice of MPNN + RNF was motivated by its linear complexity.

To make this point clearer, in this subsection, we analyze the performance of a variant of GRANOLA that uses a Subgraph GNN, namely DS-GNN [4], as the GNN$_{\text{NORM}}$, instead of an MPNN + RNF. We denote this variant as GRANOLA-SubgraphGNN. Table 6 shows that GRANOLA-SubgraphGNN behaves similarly to GRANOLA. However, employing a SubgraphGNN as the GNN$_{\text{NORM}}$ results in additional complexity coming from the Subgraph GNN itself (which is quadratic rather than linear). Thus, while enhanced expressivity in GNN$_{\text{NORM}}$ does not require employing RNF specifically, using an MPNN + RNF provides the advantage of linear complexity.

Table 6: Using RNF vs. Subgraph GNN for Expressiveness in GRANOLA. While alternative expressive GNNs for GNN$_{\text{NORM}}$ result in comparable performance, the MPNN + RNF approach offers the benefit of retaining linear complexity.

| Method | ZINC MAE ↓ | MOLHIV ROC-AUC ↑ |
|---|---|---|
| GIN + GRANOLA-SubgraphGNN | 0.1186±0.008 | 78.62±1.31 |
| GIN + GRANOLA (Using RNF) | 0.1203±0.006 | 78.98±1.17 |

Table 7: A comparison where we further modified BatchNorm to be graph adaptive using our GRANOLA design. GRANOLA-BatchNorm surpasses BatchNorm, showcasing the importance of graph adaptivity across different normalization blueprints.

| Method | ZINC-12K ↓ | MOLHIV ↑ |
|---|---|---|
| GIN + BatchNorm | $0.1630 \pm 0.004$ | $75.58 \pm 1.40$ |
| GIN + LayerNorm-node | $0.1649 \pm 0.009$ | $75.24 \pm 1.71$ |
| GIN + GRANOLA-BatchNorm | $0.1397 \pm 0.007$ | $77.93 \pm 1.22$ |
| GIN + GRANOLA (LayerNorm-node) | $0.1203 \pm 0.006$ | $78.98 \pm 1.17$ |

### H.3 GRANOLA building from BatchNorm

As discussed in Section 3, in Equation (10), $\mu_{b,n,c}$ and $\sigma_{b,n,c}$ are the mean and std of $\tilde{\mathbf{H}}^{(\ell)}$ computed per node across the feature dimension, exactly as in LayerNorm-node. This choice was dictated by the fact that we found LayerNorm-node to offer consistent performance across different benchmarks. However, it is also possible to compute $\mu_{b,n,c}$ and $\sigma_{b,n,c}$ across different dimensions, resulting in a variant of GRANOLA that builds on top of other normalization layers rather than LayerNorm-node. In Table 7 we study the effectiveness of the variant of GRANOLA that builds on top of BatchNorm. As can be seen from the table, GRANOLA-BatchNorm outperforms BatchNorm, further highlighting the role of graph adaptivity in normalizations.

### H.4 Shared weights vs. Per Layer GNN$_{\text{NORM}}^{(\ell)}$

In our experiments in Section 5, the additional normalization GNN layer, GNN$_{\text{NORM}}^{(\ell)}$, had unique parameters per layer, as evidenced by the superscript $\ell$. In Table 8, we show that by using the same GNN$_{\text{NORM}}^{(\ell)}$ for all $\ell$, that is by sharing the weights across all layers, which overall requires less parameters, our GRANOLA continues to offer competitive results, further highlighting its competitiveness also in cases where parameter budget is low.

### H.5 The role of GRANOLA Normalization

To better understand the contribution of GRANOLA, we provide results where we set the normalization term in GRANOLA to zero, leaving only the bias term to be learned. That is achieved by setting $\gamma_{b,n,c}^{(\ell)} = 0$ in Equation (10). By following this approach, we isolate the contribution of the normalization itself from the bias in the normalization process. Our results, shown in Table 9, indicate that the normalization term in GRANOLA is significant, and cannot be replaced by a simple bias.

### H.6 RNF as PE Ablation Study

GRANOLA benefits from (i) enhanced expressiveness, and (ii) graph adaptivity. Property (i) is obtained by augmenting our normalization scheme with RNF, as shown in Figure 3. Therefore, it is important to ensure that the contribution GRANOLA does not stem solely from the use of RNF, but rather the overall approach and design of our method.

To this end, in addition to the natural baseline of RNF-PE, which uses RNF as positional encoding combined with GIN + BatchNorm (as in [56]), we now provide results of RNF-PE when combined with GIN and different normalization layers. Specifically, we consider Identity (no normalization) and LayerNorm (both graph and node variants). The results are provided in Table 10, together with

Table 8: Comparison of GRANOLA with its variant GRANOLA-SHAREDGNN$_{\text{NORM}}$ obtained by sharing GNN$_{\text{NORM}}^{(\ell)}$ across layers (instead of having a different one for each layer), and BatchNorm and LayerNorm-node for reference. While using a GNN$_{\text{NORM}}^{(\ell)}$ per GNN layer leads to better results, sharing it for all $\ell$ also offers significant improvements over baseline methods.

| Method | ZINC-12K $\downarrow$ | MOLHIV $\uparrow$ |
|---|---|---|
| GIN + BatchNorm | $0.1630 \pm 0.004$ | $75.58 \pm 1.40$ |
| GIN + LayerNorm-node | $0.1649 \pm 0.009$ | $75.24 \pm 1.71$ |
| GIN + GRANOLA-SHAREDGNN$_{\text{NORM}}$ | $0.1293 \pm 0.009$ | $78.47 \pm 1.20$ |
| GIN + GRANOLA | $0.1203 \pm 0.006$ | $78.98 \pm 1.17$ |

Table 9: Comparison of GRANOLA with its variant where $\gamma_{b,n,c}^{(\ell)} = 0$ in Equation (10) shows the importance of the normalization in GRANOLA.

| Method | ZINC-12K $\downarrow$ | MOLHIV $\uparrow$ |
|---|---|---|
| GIN + GRANOLA-$\beta_{b,n,c}^{(\ell)}$-only | $0.1928 \pm 0.018$ | $74.11 \pm 1.39$ |
| GIN + GRANOLA | $0.1203 \pm 0.006$ | $78.98 \pm 1.17$ |

the results of our GRANOLA, for reference and convenience of comparison. The results suggest that while the different variants of RNF-PE do not show significant improvement over the baseline of GIN + BatchNorm, our GRANOLA does. These results are further evidence that while RNF are theoretically powerful, they may not be significant in practice, as shown in Eliasof et al. [22]. Instead, it is important to incorporate them in a thoughtful manner, for example, to obtain graph adaptivity within the normalization layer, as in our GRANOLA.

### H.7 Normalization GNN depth ablation study

As discussed in Section 3, our GRANOLA utilizes a GNN to learn graph-adaptive normalization shift and scaling parameters, and we denote this GNN by GNN$_{\text{NORM}}^{(\ell)}$. Combined with the RNF as part of the input to GNN$_{\text{NORM}}^{(\ell)}$, we are able to obtain both enhanced expressiveness (from RNF) and graph-adaptivity (by GNN$_{\text{NORM}}^{(\ell)}$). It is therefore interesting to study the effect of the number of layers in GNN$_{\text{NORM}}^{(\ell)}$ on the downstream performance. Specifically, in the case where GNN$_{\text{NORM}}^{(\ell)}$ has 0 layers, the experiment defaults to a model very similar the RNF-NORM baseline (with the difference that in RNF-NORM we have $\mathbf{Z}_b^{(\ell)} = \mathbf{R}_b^{(\ell)}$, while in this case we have $\mathbf{Z}_b^{(\ell)} = \tilde{\mathbf{H}}_b^{(\ell)} \oplus \mathbf{R}_b^{(\ell)}$), thereby losing graph adaptivity. In Table 11, we provide results on a varying number of layers, from 0 to 4. Our results suggest that there is a significant importance in terms of performance to having graph adaptivity in the normalization technique, as offered by our GRANOLA.

### H.8 Comparison with additional baselines

In the main paper, in Section 5, we focused on providing a comprehensive comparison with directly comparable methods, i.e., standard normalization methods, graph normalization methods, as well as our own set of natural baselines. In this section, we provide additional comparisons with other expressive approaches, such as positional encoding methods and Subgraph GNNs. Our additional comparisons on ZINC-12K, OGB, and TUDatasets are provided in Table 12, Table 13, and Table 14, respectively.

It is important to note, that while some of these methods achieve better performance than our GRANOLA, they are not within the same complexity class as GRANOLA, and specifically, they are not linear with respect to the number of nodes and edges in the graph, as discussed in Appendix G. For example, RFP-QR-$\hat{\mathbf{L}}, \hat{\mathbf{A}}, \mathbf{S}^{\text{learn}}$-DSS [22], which also utilizes the expressive power of RNF, achieves an MAE of 0.1106 on ZINC-12K, while our GRANOLA achieves 0.1203. However, the former is of quadratic complexity with respect to the number of nodes, while GRANOLA is linear, as standard MPNNs. On the other hand, the linear and thus directly comparable RFP - $\ell_2$ - $\hat{\mathbf{L}}, \hat{\mathbf{A}}$ achieves a higher (worse) MAE of 0.1368. Similarly, while there are Subgraph GNNs that can

Table 10: Ablation study of RNF-PE with GIN and various normalization methods.

| Method | ZINC MAE ↓ | MOLHIV ROC-AUC ↑ |
|---|---|---|
| GIN + BatchNorm [63] | 0.1630±0.04 | 75.58±1.40 |
| GIN + BatchNorm + RNF-PE [56] | 0.1621±0.014 | 75.98±1.63 |
| GIN + LayerNorm-node + RNF-PE | 0.1663±0.015 | 76.22±1.58 |
| GIN + LayerNorm-graph + RNF-PE | 0.1624±0.018 | 76.49±1.64 |
| GIN + Identity + RNF-PE | 0.2063±0.018 | 75.31±2.04 |
| GRANOLA | 0.1203±0.006 | 78.98±1.17 |

Table 11: Ablation study of the depth (number of layers) of $\text{GNN}_{\text{NORM}}^{(\ell)}$

| Depth | 0 | 1 | 2 | 3 | 4 |
|---|---|---|---|---|---|
| ZINC (MAE ↓) | 0.1562±0.013 | 0.1218±0.009 | 0.1203±0.006 | 0.1209±0.010 | 0.1224±0.008 |
| MOLHIV (ROC-AUC ↑) | 77.61±1.64 | 78.33±1.34 | 78.98±1.17 | 78.86±1.20 | 78.21±1.31 |

outperform GRANOLA, these are at least a quadratic in the number of nodes. Therefore, we find that our GRANOLA offers a practical yet powerful approach for utilizing RNF.

Additionally, we observe that in some cases, GRANOLA achieves similar or better performance than other expressive and asymptotically more complex methods. For example, our results on TUDatsets in Table 14 show that GRANOLA offers better performance than DSS-GNN [4] on all considered datasets, despite DSS being quadratic in the number of nodes.

Table 12: Additional comparisons of GRANOLA with various baselines on the ZINC-12K graph dataset. All methods obey to the 500k parameter budget.

| Method | ZINC (MAE ↓) |
|---|---|
| **MPNNs** | |
| GCN [37] | 0.321±0.009 |
| PNA [15] | 0.133±0.011 |
| **POSITIONAL ENCODING METHODS** | |
| GIN + Laplacian PE [22] | 0.1557±0.012 |
| RFP - $\ell_2$ - $\hat{\mathbf{L}}$, $\hat{\mathbf{A}}$ [22] | 0.1368±0.010 |
| RWPE [20] | 0.1279±0.005 |
| RFP-QR-$\hat{\mathbf{L}}$, $\hat{\mathbf{A}}$, $\mathbf{S}^{\text{learn}}$-DSS [22] | 0.1106±0.012 |
| **DOMAIN-AWARE GNNs** | |
| GSN [10] | 0.101±0.010 |
| CIN [8] | 0.079±0.006 |
| **HIGHER ORDER GNNs** | |
| PPGN [40] | 0.079±0.005 |
| PPGN++ (6) [52] | 0.071±0.001 |
| **GRAPH TRANSFORMERS** | |
| GPS [53] | 0.070±0.004 |
| GRAPHORMER [66] | 0.122±0.006 |
| GRAPHORMER-GD [70] | 0.081±0.009 |
| **SUBGRAPH GNNs** | |
| NGNN [71] | 0.111±0.003 |
| DS-GNN (EGO+) [4] | 0.105±0.003 |
| DSS-GNN (EGO+) [4] | 0.097±0.006 |
| GNN-AK [73] | 0.105±0.010 |
| GNN-AK+ [73] | 0.091±0.011 |
| SUN (EGO+) [25] | 0.084±0.002 |
| GNN-SSWL [69] | 0.082±0.003 |
| GNN-SSWL+ [69] | 0.070±0.005 |
| DS-GNN (NM) [5] | 0.087±0.003 |
| **NATURAL BASELINES** | |
| GIN + BatchNorm + RNF-PE [56] | 0.1621±0.014 |
| GIN + RNF-NORM | 0.1562±0.013 |
| **STANDARD NORMALIZATION LAYERS** | |
| GIN + BatchNorm [63] | 0.1630±0.004 |
| GIN + InstanceNorm [58] | 0.2984±0.017 |
| GIN + LayerNorm-node [3] | 0.1649±0.009 |
| GIN + LayerNorm-graph [3] | 0.1609±0.014 |
| GIN + Identity | 0.2209±0.018 |
| **GRAPH NORMALIZATION LAYERS** | |
| GIN + PairNorm [72] | 0.3519±0.008 |
| GIN + MeanSubtractionNorm [64] | 0.1632±0.021 |
| GIN + DiffGroupNorm [74] | 0.2705±0.024 |
| GIN + NodeNorm [75] | 0.2119±0.017 |
| GIN + GraphNorm [11] | 0.3104±0.012 |
| GIN + GraphSizeNorm [21] | 0.1931±0.016 |
| GIN + SuperNorm [12] | 0.1574±0.018 |
| GIN + GRANOLA-NO-RNF | 0.1497±0.008 |
| GIN + GRANOLA-MS | 0.1238±0.009 |
| GIN + GRANOLA | 0.1203±0.006 |

Table 13: Additional comparisons of GRANOLA to natural baselines, standard and graph normalization layers, and subgraph GNNs, demonstrating the practical advantages of our approach. – indicates the result was not reported in the original paper.

| Method ↓ / Dataset → | MOLESOL RMSE ↓ | MOLTOX21 ROC-AUC ↑ | MOLBACE ROC-AUC ↑ | MOLHIV ROC-AUC ↑ |
|---|---|---|---|---|
| **MPNNs** | | | | |
| GCN [37] | 1.114±0.036 | 75.29±0.69 | 79.15±1.44 | 76.06±0.97 |
| GIN [63] | 1.173±0.057 | 74.91±0.51 | 72.97±4.00 | 75.58±1.40 |
| **POSITIONAL ENCODING METHODS** | | | | |
| GIN + Laplacian PE [22] | – | – | – | 77.88±1.82 |
| RFP - $\ell_2$ - $\hat{\mathbf{L}}$, $\hat{\mathbf{A}}$ [22] | – | – | – | 77.91±1.43 |
| RWPE [20] | – | – | – | 78.62±1.13 |
| RFP-QR-$\hat{\mathbf{L}}$, $\hat{\mathbf{A}}$, $\mathbf{S}^{\text{learn}}$-DSS [22] | – | – | – | 80.58±1.21 |
| **EXPRESSIVE GNNS** | | | | |
| GSN [10] | – | – | – | 80.39±0.90 |
| CIN [8] | – | – | – | 80.94±0.57 |
| **SUBGRAPH GNNS** | | | | |
| RECONSTR. GNN [16] | 1.026±0.033 | 75.15±1.40 | – | 76.32±1.40 |
| NGNN [71] | – | – | – | 78.34±1.86 |
| DS-GNN (EGO+) [4] | – | 76.39±1.18 | – | 77.40±2.19 |
| DSS-GNN (EGO+) [4] | – | 77.95±0.40 | – | 76.78±1.66 |
| GNN-AK+ [73] | – | – | – | 79.61±1.19 |
| SUN (GIN) (EGO+) [25] | – | – | – | 80.03±0.55 |
| GNN-SSWL+ [69] | – | – | – | 79.58±0.35 |
| DS-GNN (NM) [5] | 0.847±0.015 | 76.25±1.12 | 78.41±1.94 | 76.54±1.37 |
| **NATURAL BASELINES** | | | | |
| GIN + BatchNorm + RNF-PE [56] | 1.052±0.041 | 75.14±0.67 | 74.28±3.80 | 75.98±1.63 |
| GIN + RNF-NORM | 1.039±0.040 | 75.12±0.92 | 77.96±4.36 | 77.61±1.64 |
| **STANDARD NORMALIZATION LAYERS** | | | | |
| GIN + BatchNorm [63] | 1.173±0.057 | 74.91±0.51 | 72.97±4.00 | 75.58±1.40 |
| GIN + InstanceNorm [58] | 1.099±0.038 | 73.82±0.96 | 74.86±3.37 | 76.88±1.93 |
| GIN + LayerNorm-node [3] | 1.058±0.024 | 74.81±0.44 | 77.12±2.70 | 75.24±1.71 |
| GIN + LayerNorm-graph [3] | 1.061±0.043 | 75.03±1.24 | 76.49±4.07 | 76.13±1.84 |
| GIN + Identity | 1.164±0.059 | 73.34±1.08 | 72.55±2.98 | 71.89±1.32 |
| **GRAPH NORMALIZATION LAYERS** | | | | |
| GIN + PairNorm [72] | 1.084±0.031 | 73.27±1.05 | 75.11±4.24 | 76.18±1.47 |
| GIN + MeanSubtractionNorm [64] | 1.062±0.045 | 74.98±0.62 | 76.36±4.47 | 76.37±1.40 |
| GIN + DiffGroupNorm [74] | 1.087±0.063 | 74.48±0.76 | 75.96±3.79 | 74.37±1.68 |
| GIN + NodeNorm [75] | 1.068±0.029 | 73.27±0.83 | 75.67±4.03 | 75.50±1.32 |
| GIN + GraphNorm [11] | 1.044±0.027 | 73.54±0.80 | 73.23±3.88 | 78.08±1.16 |
| GIN + GraphSizeNorm [21] | 1.121±0.051 | 74.07±0.30 | 76.18±3.52 | 75.44±1.51 |
| GIN + SuperNorm [12] | 1.037±0.044 | 75.08±0.98 | 75.12±3.38 | 76.55±1.76 |
| GIN + GRANOLA-NO-RNF | 1.088±0.032 | 75.87±0.72 | 76.23±2.06 | 77.09±1.49 |
| GIN + GRANOLA-MS | 0.971±0.026 | 77.32±0.67 | 79.18±2.41 | 78.84±1.22 |
| GIN + GRANOLA | 0.960±0.020 | 77.19±0.85 | 79.92±2.56 | 78.98±1.17 |

Table 14: Additional Graph classification accuracy (%) ↑ on TUDatasets. – indicates the result was not reported in the original paper.

| Method ↓ / Dataset → | MUTAG | PTC | PROTEINS | NCI1 | NCI109 |
|---|---|---|---|---|---|
| **EXPRESSIVE GNNs** | | | | | |
| GSN [10] | 92.2±7.5 | 68.2±7.2 | 76.6±5.0 | 83.5±2.0 | – |
| SIN [9] | – | – | 76.4±3.3 | 82.7±2.1 | – |
| CIN [8] | 92.7±6.1 | 68.2±5.6 | 77.0±4.3 | 83.6±1.4 | 84.0±1.6 |
| **SUBGRAPH GNNs** | | | | | |
| DROPEDGE [54] | 91.0±5.7 | 64.5±2.6 | 73.5±4.5 | 82.0±2.6 | 82.2±1.4 |
| GRAPHCONV + ID-GNN [67] | 89.4±4.1 | 65.4±7.1 | 71.9±4.6 | 83.4±2.4 | 82.9±1.2 |
| DS-GNN (GIN) (EGO+) [4] | 91.0±4.8 | 68.7±7.0 | 76.7±4.4 | 82.0±1.4 | 80.3±0.9 |
| DSS-GNN (GIN) (EGO+) [4] | 91.1±7.0 | 69.2±6.5 | 75.9±4.3 | 83.7±1.8 | 82.8±1.2 |
| GNN-AK+ [73] | 91.3±7.0 | 67.8±8.8 | 77.1±5.7 | 85.0±2.0 | – |
| SUN (GIN) (EGO+) [25] | 92.1±5.8 | 67.6±5.5 | 76.1±5.1 | 84.2±1.5 | 83.1±1.0 |
| **NATURAL BASELINES** | | | | | |
| GIN + BatchNorm + RNF-PE [56] | 90.8±4.8 | 64.4±6.7 | 74.1±2.6 | 82.1±1.5 | 81.3±1.1 |
| GIN + RNF-NORM | 88.9±5.1 | 67.1±4.3 | 76.4±4.8 | 81.8±2.2 | 81.9±2.5 |
| **STANDARD NORMALIZATION LAYERS** | | | | | |
| GIN + BatchNorm [63] | 89.4±5.6 | 64.6±7.0 | 76.2±2.8 | 82.7±1.7 | 82.2±1.6 |
| GIN + InstanceNorm [58] | 90.5±7.8 | 64.7±5.9 | 76.5±3.9 | 81.2±1.8 | 81.8±1.6 |
| GIN + LayerNorm-node [3] | 90.1±5.9 | 65.3±4.7 | 76.2±3.0 | 81.9±1.5 | 82.0±2.1 |
| GIN + Layernorm-graph [3] | 90.4±6.1 | 66.4±6.5 | 76.1±4.9 | 82.0±1.6 | 81.5±1.3 |
| GIN + Identity | 87.9±7.8 | 63.1±7.2 | 75.8±6.3 | 81.3±2.1 | 80.6±1.7 |
| **GRAPH NORMALIZATION LAYERS** | | | | | |
| GIN + PairNorm [72] | 87.8±7.1 | 67.1±6.3 | 76.7±4.8 | 75.8±2.1 | 75.3±1.4 |
| GIN + MeanSubtractionNorm [64] | 90.1±5.4 | 68.0±5.9 | 76.4±4.6 | 79.2±1.2 | 79.0±1.1 |
| GIN + DiffGroupNorm [74] | 87.8±7.6 | 67.4±6.8 | 76.9±4.3 | 77.2±2.6 | 77.1±1.9 |
| GIN + NodeNorm [75] | 88.3±7.0 | 65.1±8.3 | 74.5±4.6 | 81.2±1.4 | 79.4±1.0 |
| GIN + GraphNorm [11] | 91.6±6.5 | 64.9±7.5 | 77.4±4.9 | 81.4±2.4 | 82.4±1.7 |
| GIN + GraphSizeNorm [21] | 88.2±6.3 | 68.0±8.1 | 77.0±5.0 | 79.8±1.5 | 80.1±1.8 |
| GIN + SuperNorm [12] | 89.3±5.6 | 64.7±3.9 | 76.1±4.7 | 83.0±1.5 | 82.8±1.7 |
| GIN + GRANOLA-NO-RNF | 89.7±5.4 | 65.8±5.7 | 76.6±2.5 | 83.1±1.2 | 83.0±1.5 |
| GIN + GRANOLA-MS | 92.1±4.8 | 69.8±4.7 | 77.3±3.5 | 84.3±1.5 | 83.5±1.8 |
| GIN + GRANOLA | 92.2±4.6 | 69.9±4.5 | 77.5±3.7 | 84.0±1.7 | 83.7±1.6 |

