# OpenReview forum: "GRANOLA: Adaptive Normalization for Graph Neural Networks"
_NeurIPS.cc/2024/Conference — NeurIPS 2024 poster_

### Official Review · Reviewer_PwFk · 2024-07-05

**Soundness:** 3
**Presentation:** 3
**Contribution:** 3
**Rating:** 6
**Confidence:** 3

**Summary:**

The paper introduces a novel graph-adaptive normalization layer named GRANOLA for Graph Neural Networks (GNNs). The authors argue that existing normalization techniques, such as BatchNorm and InstanceNorm, are not well-suited for GNNs due to their design not considering the unique characteristics of graph-structured data. To address this, GRANOLA is proposed to normalize node features by adapting to the specific characteristics of the graph, using Random Node Features (RNF) to generate expressive node representations. The paper provides theoretical results supporting the design choices and demonstrates through empirical evaluation that GRANOLA outperforms existing normalization techniques across various graph benchmarks.

**Strengths:**

* The paper presents a novel normalization technique specifically tailored for GNNs, addressing a recognized gap in the field where traditional normalization layers do not capture the unique properties of graph data effectively.

* The authors provide a solid theoretical basis for their method, including proofs that GRANOLA can achieve full adaptivity to the input graph, which is a significant contribution to the understanding of normalization in GNNs.

* The paper offers extensive empirical results across multiple datasets and tasks, showing consistent performance improvements of GRANOLA over existing methods, which strengthens the credibility of the proposed technique.

**Weaknesses:**

* The work focuses on the normalization of GNNs for graph-level tasks. However, there is a type of model that dominates this field: Graph Transformers. I'm curious if this normalization would help with the graph transformers in graph-level tasks.

* Implementation details are not so clear. Is the RNF sampled once and trained like other parameters or just randomly sampled at each mini-batch (or each epoch)?

**Questions:**

See weaknesses

**Limitations:**

Yes

---

> ### Author Rebuttal · Authors · 2024-08-07
>
> We thank the reviewer for their feedback, and we are glad to see they have recognized the significance of our contribution. They have nonetheless asked some questions, which we address below.
>
> **Q1:** The work focuses on the normalization of GNNs for graph-level tasks. However, there is a type of model that dominates this field: Graph Transformers. I'm curious if this normalization would help with the graph transformers in graph-level tasks.
>
> **A1:** Following the reviewer’s suggestion, we have conducted an additional experiment and coupled GRANOLA with the GPS graph transformer (Rampášek et al., 2022), as reported in the following table.
>
> As can be seen from the table, GRANOLA improves the performance of the GPS transformer, further highlighting its versatility and ability to enhance the performance of various and diverse graph models.
>
> | Method             | ZINC-12k $\downarrow$   | OGBG-MOLHIV $\uparrow$ |
> |--------------------|-------------|-------------|
> | GPS                | 0.070±0.004 | 78.80±1.01  |
> | GPS+GRANOLA (Ours) | 0.062±0.006 | 79.21±1.26  |
>
> Rampášek et al., 2022. Recipe for a General, Powerful, Scalable Graph Transformer. NeurIPS 2022
>
> **Q2:** Implementation details are not so clear. Is the RNF sampled once and trained like other parameters or just randomly sampled at each mini-batch (or each epoch)?
>
> **A2:** We follow the standard practice of RNF methods (Abboud et al., 2021, Sato et al., 2021), and sample RNF for each mini-batch. We thank the reviewer for pointing it out and we will make it clearer in the next paper revision.
>
> Abboud et al., 2021. The Surprising Power of Graph Neural Networks with Random Node Initialization. IJCAI 2021
>
> Sato et al., 2021. Random Features Strengthen Graph Neural Networks. SDM 2021
>
> ***
>
> We are thankful to the reviewer for their constructive feedback. We made efforts to conduct the additional experiments they suggested. If they find our responses satisfactory, we kindly ask them to reconsider their rating.

---

> > ### Comment · Reviewer_PwFk · 2024-08-11
> >
> > Thanks for the authors' responses. I'll raise my score to 6

---

### Official Review · Reviewer_tP2u · 2024-07-12

**Soundness:** 3
**Presentation:** 3
**Contribution:** 3
**Rating:** 6
**Confidence:** 3

**Summary:**

This paper introduces GRANOLA (Graph Adaptive Normalization Layer), a novel normalization technique designed specifically for Graph Neural Networks (GNNs). The authors identify a critical gap in existing normalization methods for GNNs, which often fail to capture the unique structural characteristics of graph data or consistently improve performance across various tasks. GRANOLA aims to address these limitations by dynamically adjusting node features based on both the graph structure and Random Node Features (RNF). The key innovation lies in its use of an additional GNN (termed GNN_NORM) to generate normalization parameters that are adaptive to the input graph. The paper provides a comprehensive theoretical analysis of GRANOLA, demonstrating its ability to default to RNF-augmented Message Passing Neural Networks (MPNNs) and proving its increased expressive power compared to standard MPNNs. The authors also show that the use of RNF is necessary for this increased expressiveness. Empirically, the paper presents extensive evaluations across multiple graph benchmarks.

**Strengths:**

GRANOLA presents a novel approach to graph normalization by incorporating graph adaptivity and RNF, addressing a significant gap in existing methods.

The paper provides solid theoretical analysis, including proofs of GRANOLA's expressive power and its ability to default to RNF-augmented MPNNs.

The authors conduct extensive experiments across multiple datasets and tasks, demonstrating GRANOLA's consistent performance improvements over existing normalization techniques.

The paper effectively demonstrates how GRANOLA serves as a valuable bridge between the theoretical expressiveness of RNF-augmented MPNNs and practical performance improvements in graph learning tasks.

GRANOLA maintains the same asymptotic complexity as standard MPNNs while offering improved performance, making it a practical solution for real-world applications.

**Weaknesses:**

GRANOLA's parameters (gamma and beta) are unique for each node, layer, and feature attribute. This fine-grained adjustment is significantly more detailed than most normalization methods, making GRANOLA resemble a new model architecture rather than a normalization technique. The motivation as a normalization method may be somewhat inappropriate given this level of detail.

The paper could benefit from additional experiments, such as removing the $\gamma_{b,n,c}^{(\ell)}$ term and retaining only the affine intercept term $\beta_{b,n,c}^{(\ell)}$. This would help determine whether the improved performance is due to the detailed internal model adjustments or the normalization concept itself.

**Questions:**

The authors address some limitations of their work in the conclusion section. They could have elaborated more on the practical limitations of GRANOLA, such as potential challenges in implementing it in resource-constrained environments or its scalability to extremely large graphs. While the normalization GNN depth ablation study is available in Appendix H.2, a more detailed discussion on the choice of other hyperparameters for the normalization GNN, such as dimensions, would be beneficial. This is particularly important given that the normalization GNN is at the core of the GRANOLA framework.

---

> ### Author Rebuttal · Authors · 2024-08-07
>
> We are happy to see that the reviewer has appreciated the novelty of our approach, while finding our theoretical analysis solid and our empirical evaluation extensive. We would like to thank them for raising interesting points of discussion, which we address in the following.
>
> **Q1:** GRANOLA's parameters (gamma and beta) are unique for each node, layer, and feature attribute. This fine-grained adjustment is significantly more detailed than most normalization methods, making GRANOLA resemble a new model architecture rather than a normalization technique. The motivation as a normalization method may be somewhat inappropriate given this level of detail.
>
> **A1:** We agree with the reviewer that the gamma and beta are more refined than other standard normalization layers, but this level of detail is intrinsic to adaptive normalization layers. In an adaptive normalization, instead of using the same affine parameters $\gamma_{c}^{(\ell)}$ and $\beta_{c}^{(\ell)}$ (Equation (3)) for all the nodes in all the graphs, the normalization method utilizes specific parameters conditioned on the input graph. This adaptivity has proven to be a valuable property in other domains (for instance, Huang et al., 2017), and in our case is achieved by generating the affine parameters through the normalization GNN.
>
> Huang et al., 2017. Arbitrary style transfer in real-time with adaptive instance normalization. ICCV 2017
>
>
> **Q2:** The paper could benefit from additional experiments, such as removing the $\gamma_{b,n,c}^{(\ell)}$ term and retaining only the affine intercept term $\beta_{b,n,c}^{(\ell)}$. This would help determine whether the improved performance is due to the detailed internal model adjustments or the normalization concept itself.
>
>
> **A2:** Following your suggestion, we have included an additional experiment obtained by removing the $\gamma_{b,n,c}^{(\ell)}$ term by setting it to zero, while retaining only $\beta_{b,n,c}^{(\ell)}$. The following table shows that GRANOLA outperforms this method, highlighting the importance of the normalization concept.
>
> | Method                        | ZINC-12k  $\downarrow$  | OGBG-MOLHIV $\uparrow$ |
> |-------------------------------|--------------|-------------|
> | GIN+GRANOLA-$\beta_{b,n,c}^{(\ell)}$-only    | 0.1928±0.018 | 74.11±1.39  |
> | GIN+GRANOLA (As in the paper) | 0.1203±0.006 | 78.98±1.17  |
>
> **Q3:** The authors address some limitations of their work in the conclusion section. They could have elaborated more on the practical limitations of GRANOLA, such as potential challenges in implementing it in resource-constrained environments or its scalability to extremely large graphs.
>
> **A3:** Thank you for the suggestion. We agree that expanding on the practical limitations of GRANOLA is valuable. In practice, adding GRANOLA to standard MPNNs does not significantly impact memory usage (and it maintains the linear space complexity of MPNNs). Therefore, GRANOLA faces the same challenges of existing and widely-used GNN methods when dealing with extremely large graphs. We will elaborate on this point in the next revision.
>
> **Q4:** While the normalization GNN depth ablation study is available in Appendix H.2, a more detailed discussion on the choice of other hyperparameters for the normalization GNN, such as dimensions, would be beneficial. This is particularly important given that the normalization GNN is at the core of the GRANOLA framework.
>
> **A4:** Following the reviewer's suggestion, we will include a more thorough discussion of the hyperparameters for our $\text{GNN}\_{NORM}$. For completeness, we note that $\text{GNN}\_{NORM}$ maintains the same embedding dimensions as the outer GNN layer, and our hyperparameter search includes different choices for the number of layers within $\text{GNN}\_{NORM}$. While other configurations are possible, we found these choices to be effective in practice and help reduce the number of parameter choices.
>
> ***
>
> We are grateful to the reviewer for their feedback. We made efforts to address all the concerns raised, and, if they feel the same, we kindly ask them to reconsider their rating.

---

> > ### Comment · Reviewer_tP2u · 2024-08-13
> >
> > Thank you for the detailed response. I raise the score to 6.

---

### Official Review · Reviewer_Ao3Y · 2024-07-12

**Soundness:** 3
**Presentation:** 3
**Contribution:** 2
**Rating:** 6
**Confidence:** 5

**Summary:**

The paper pertains to the problem of using normalisation techniques specifically designed for graph-structured data and Graph Neural Networks (GNNs). Using constructed illustrative examples, the authors claim that existing normalisation techniques (including others designed for graphs) may create expressivity issues to the GNNs to which they are applied. Motivated by these examples, they speculate that one possible reason is the lack of adaptivity of the normalisation (affine) parameters to the input graph. To that end, they propose to replace fixed normalisation parameters with adaptive ones, produced at each layer by an auxiliary GNN for each vertex of the input graph. Additionally (and probably crucially), the auxiliary GNNs do not only receive the current vertex features as inputs but also Random Node Features (typically sampled from a Gaussian). The last one is known to render GNNs universal, and so is the case here. Experimentally, the method is tested on a battery of tasks and ablated across several factors (training convergence speed, combined with more expressive architectures than plain MPNNs), showing consistently improved performance compared to other normalisation methods and competitiveness with the state-of-the-art in several cases.

**Strengths:**

**Clarity**. The paper is well-written, easy-to-follow and all the concepts are clearly explained. It can therefore be understood by a wider audience.

**Contextualisation to related work**. An extensive review and comparisons with related works are provided, allowing the reader to grasp their differences and understand the innovation of the proposed approach.

**Importance to the GNN community.** The techniques proposed by the authors are of general interest to the GNN community since they clearly and consistently improve current normalisation techniques and show good potential for adaptation in practice in various problems. Additionally, it is one of the few works showcasing the practical benefits of RNF (however see also weaknesses), which is a long-standing puzzle for GNN researchers.

**Empirical results**. The proposed method appears to work quite well and consistently in practice, in terms of empirical generalisation. In particular, it provides improvements against all base architectures tested and against all competing normalisation techniques, while both methodological techniques proposed (adaptive normalisation + RNF) are sufficiently ablated and shown to be important to be combined to obtain the desired performance gains.

**Weaknesses:**

**Reservations regarding claims and the selected way to present findings**.
- One of my main objections to the presented manuscript is the way the findings were selected to be presented. In particular, the authors present their work as a graph-oriented adaptive normalisation technique and motivate their approach by comparing it against other normalisation techniques, both intuitively using constructed counterexamples and empirically, showcasing consistent improvements. Nevertheless, it seems to me that the main reason behind the empirical success is the random features. For example, see Table 2, where the adaptive normalisation per se does not seem to provide any significant practical advantages.
- However, as the authors correctly point out, random features are known to behave subpar in practice, despite their theoretical advantages. This leads me to believe that the authors have probably found a way to *improve the performance of random features*. To me, this is an important contribution per se, but the authors have chosen not to present their paper in that way, nor to sufficiently point out this takeaway in their paper.
- On the other hand, the authors have devoted a substantial part of the paper to motivating adaptive normalisation and discussing other normalisation techniques. In my viewpoint, the intuitive explanations provided (the examples in sec. 2.2. and the last paragraph before section 3.1.) are speculative and do not seem to be the real reason behind the success of the approach.
- Note also, that in several parts of the paper, the authors claim that existing normalisation techniques limit MPNNs’ expressivity, but I am not sure if this is the case. I think that if the authors choose to keep these claims, they should probably be provided with a more rigorous theoretical statement.
- I think that the most important question that needs to be addressed is why incorporating RNF into a normalisation layer overcomes their current limitations in providing performance improvements. I recommend that the authors discuss this both in their rebuttal and the paper, as it will provide important insights.

**Efficiency**. Another reservation I have about this work is that computational efficiency might be an issue, but is not adequately discussed.  For example, the runtimes provided by the authors in Table 4 in the appendix, show an almost 3-fold increase in both training and inference time. Although this might not be significant compared to other more expressive GNNs, the comparison between the performance-complexity trade-offs is not clear. Moreover, it is a limitation of this method and should be more clearly discussed in the main paper.

**Limited evaluation against baselines**. Although as I mentioned before, the results are indeed convincing, both from a perspective of a normalisation technique and of a random feature technique, the authors have not sufficiently compared against the state-of-the-art. To their credit, they did include other baselines in the supplementary material and mentioned the gap in performance in their limitations section, but I think it would be fairer to discuss this more prominently (especially since this work can be perceived as an expressive GNN and not as a normalisation technique alone).

**Limited technical novelty**: Finally, a minor weakness is that there is limited novelty from a technical perspective, since if I am not mistaken adaptive normalisation has been proposed before for other domains (e.g. AdaIN, Huang et al., ICCV’17 is a relevant example – discussed by the authors in the appendix) and random features are well-known in the GNN community.

**Questions:**

- Have the authors tried to modify a different normalisation technique, apart from LayerNorm? This might be an interesting ablation study.
- Do the authors need a GNN normalisation network for each layer of the GNN processing network?

**Limitations:**

The authors have mentioned some limitations (mainly the gap in performance compared to state-of-the-art), but I think others should be discussed more extensively (e.g. efficiency - see weaknesses).

I do not foresee any negative societal impact.

---

> ### Author Rebuttal · Authors · 2024-08-07
>
> We thank the reviewer for the feedback, and the recognition of the importance of this work for the GNN community. The reviewer also raised concerns, which we address below.
>
> **Q1:** ..it seems to me that the main reason behind the empirical success is the random features.
>
> **A1:** We found that the combination of RNF and graph adaptivity drives GRANOLA's success, as shown by the worse performance of RNF-PE and RNF-NORM (lines 263-266), which lack graph adaptivity, and GRANOLA-NO-RNF, which omits RNF, compared to GRANOLA.
>
> **Q2:** ..This leads me to believe that the authors have probably found a way to improve the performance of random features. To me, this is an important contribution per se.
>
> **A2:**  We agree that our method makes RNF practical and, thus, offering a contribution to RNF research. We discussed this in the paper (lines 235-245),  and we will make it more prominent.
>
> We remark that other expressive GNNs can be used for the normalization function (lines 245-249). Thus, we have conducted an experiment, using DS-GNN [1] as our $\text{GNN}\_\text{NORM}$, instead of an MPNN + RNF.
> The table shows that GRANOLA-SubgraphGNN behaves similarly to GRANOLA,  but with additional complexity of the Subgraph GNN. This shows that the expressivity of $\text{GNN}_\text{NORM}$ does not necessarily need to be achieved by RNF, and our choice was motivated by the linear complexity of MPNNs + RNF.
>
> | | ZINC-12k $\downarrow$ | OGBG-MOLHIV $\uparrow$ |
> |--|--|--|
> | GIN+GRANOLA-SubgraphGNN | 0.1186±0.008 | 78.62±1.31  |
> | GIN+GRANOLA (Using RNF, as in the paper) | 0.1203±0.006 | 78.98±1.17 |
>
> [1] Bevilacqua et al., 2022. Equivariant Subgraph Aggregation Networks
>
> **Q3:** In several parts of the paper, the authors claim that existing normalisation techniques limit MPNNs’ expressivity, but I am not sure if this is the case.
>
> **A3:** This result is explained in Cai et al., 2021, and we provide a slightly more formal explanation, which we will expand in the paper.
>
> Theorem: Let $f$ be a stacking of GIN layers with non-linear activations followed by sum pooling. Let $f^{\text{norm}}$ be the architecture obtained by adding InstanceNorm or BatchNorm without affine parameters. Then $f^{\text{norm}}$ is strictly less expressive than $f$.
>
> Proof Sketch: All non-isomorphic graphs that can be distinguished by $f^{\text{norm}}$ can clearly be distinguished by $f$. To show that $f^{\text{norm}}$ is strictly less expressive than $f$, consider two CSL graphs with different numbers of nodes. These are distinguishable by $f$. However, applying InstanceNorm to the output of GIN results in a zero matrix (Proposition 4.1 in Cai et al., 2021). Similarly, if the batch consists of these two graphs, applying BatchNorm results in a zero matrix. Since the output of the normalization is a zero matrix, they are indistinguishable by $f^{\text{norm}}$.
>
> Cai et al, 2021. GraphNorm: A Principled Approach to Accelerating Graph Neural Network Training
>
> **Q4:** Why incorporating RNF into a normalisation layer overcomes their current limitations in providing performance improvements.
>
> **A4:** We agree with the reviewer that this corresponds to our main research question, and we will add a more thorough discussion.
>
> Incorporating RNF into our GRANOLA allows it to *fully adapt to the input graph*,  providing different affine parameters for non-isomorphic nodes. Full adaptivity is lost when removing RNF and using a standard MPNN as $\text{GNN}\_\text{NORM}$, as in GRANOLA-no-RNF. This is because GRANOLA-no-RNF is not more expressive than an MPNN (Prop. 4.1), and thus, there exist non-isomorphic nodes that will get the same representation (and the same affine parameters).  However, any other most expressive architecture used as $\text{GNN}_\text{NORM}$ would achieve the same full adaptivity, and our choice of MPNN + RNF was motivated by its linear complexity.
>
> **Q5:** Another reservation I have about this work is that computational efficiency might be an issue, but is not adequately discussed.
>
> **A5:** We agree that the performance improvement of GRANOLA requires additional computations (Appendix G). However, the overall runtime remains a fraction of more expressive methods (5.2x faster than efficient expressive models like SubgraphGNNs).
>
> We remark that practitioners often have different needs from their models. At times, they might prefer accuracy over efficiency (e.g., drug discovery). In such scenarios, GRANOLA offers a strong tradeoff between performance and cost. We will clarify this in the revision.
>
> **Q6:** … adaptive normalisation has been proposed before for other domains (e.g. AdaIN, Huang et al., ICCV’17 is a relevant example – discussed by the authors in the appendix) and random features are well-known in the GNN community.
>
> **A6:** Despite some similarities, GRANOLA differs significantly from AdaIN, which adjusts the content's mean and variance to match the *style input*. Additionally, GRANOLA employs RNF for full graph adaptivity, but other expressive methods could also be used as shown in A2.
>
> **Q7:** Have the authors tried to modify a different normalisation technique, apart from LayerNorm?
>
> **A7:** Thank you for the suggestion. We modified BatchNorm to be graph adaptive using our GRANOLA design. Table 1 in the additional PDF shows that GRANOLA-BatchNorm outperforms BatchNorm, highlighting the role of graph adaptivity in normalizations.
>
> **Q8:**  Do the authors need a GNN normalisation network for each layer of the GNN processing network?
>
> **A8:** We use a shallow GNN normalization network for each layer. We also tested a variant with a shared normalization GNN across all layers. Table 2 in the additional PDF shows that while per-layer normalization GNNs yield better results, sharing one still significantly improves over the baselines
>
> ***
> We sincerely appreciate the reviewer’s feedback. We have made efforts to address their questions thoroughly, and if they feel the same, we kindly ask them to reconsider their rating.

---

> ### Comment · Reviewer_Ao3Y · 2024-08-12
> **Post-rebuttal**
>
> I thank the authors for their response and the additional experiments provided. As I mentioned in my initial review, combining adaptive normalisation with random features works well in practice, but Ι still find the underlying reasons behind this success unclear.
>
> The authors provided in their rebuttal an additional experiment, replacing random features with Subgraph-GNNs (as an alternative to boost the expressivity of the function computing the normalisation parameters) obtaining similar results. Although this hints that adaptive normalisation can be successful with any technique that improves expressivity, it triggers an additional question: should we credit success to extra expressivity or adaptive normalisation? At this point, it should be noted that, if I am not mistaken, using Subgraph-GNNs alone performs better than GIN+GRANOLA-SubgraphGNN.
>
> In any case, the overall picture remains a bit blurry: the authors put significant focus during their presentation on the adaptive normalisation perspective of their approach, yet this works well only when combined with more expressive functions computing the normalisation parameters. However, this, in turn, might be outperformed by more expressive architectures alone. I agree with the argument that the latter is not the case for random features, which brings me back to my initial observation that the authors have found a way to exploit the theoretical advantages of random features (along with their linear complexity). However, the rest of the claims need further work to be made more convincing.
>
> Overall, my reservations do not concern the method (the results are convincing), but rather the explanations/insights provided w.r.t. its behaviour. In other words, I think the paper deserves attention due to its empirical findings, but to some extent lacks maturity. Therefore, I will keep my initial score and recommendation.

---

> ### Author Response · Authors · 2024-08-12
> **Post-rebuttal follow up**
>
> **We thank the Reviewer for the discussion and the added comments.  We are also happy to read that the Reviewer finds our results convincing and deserving attention.**
>
> **We would like to reply to your important comments one by one. We hope that you find them satisfactory, and we would like to receive your feedback. We implemented your comments and their follow-up discussions to the revised paper.**
>
> ***
>
> **Regarding Subgraph-GNNs vs. GRANOLA:** The reviewer rightfully mentions that there are Subgraph-GNNs that can outperform GRANOLA, and this is also reflected in our results in Appendix in Tables 7, 8, and 9. **However**, it is important to note that our paper does not make claims about outperforming subgraph GNNs. Rather, it **focuses on GNNs with linear time complexity**
> Importantly, this experiment aims to demonstrate that effective graph normalization requires both **adaptivity** and **expressivity**. Our approaches achieve this combination using random features, which allows for implementation with linear time complexity.
> Furthermore, please note that in the added  experiment following your questions, we used a DS-GNN [D1] with Ego-network subgraphs. This approach does not yield higher results on the tested datasets compared with our GRANOLA, as we show in the Table below:
>
> | Method                  | ZINC-12k $\downarrow$ | OGBG-MOLHIV $\uparrow$ |
> |-------------------------|-------------------------|--------------------------|
> | DS-GNN (GIN) (Ego)      | 0.126±0.006             | 78.00±1.42               |
> | GIN+GRANOLA-SubgraphGNN | **0.1186±0.008**            |  **78.62±1.31**               |
>
>
> Therefore, it is important to note that, despite us not claiming that GRANOLA is meant to compete with subgraph GNNs, it does offer better performance than the core subgraph GNN used in our added experiment. We added this important discussion to our revised paper.
>
> [D1] Bevilacqua et al., 2022. Equivariant Subgraph Aggregation Networks.
>
> ***
>
>
> **Regarding expressivity vs adaptivity:** Thank you for the comment. We kindly note that in our paper, we highlight the discussion of the desire to have **both expressivity and adaptivity** several times, and we verify these claims by extensive experiments.
>
> *We now list the discussions on this in the paper:*
>
> 1. Lines 35-40: We mention that full adaptivity can be obtained by combining an adaptive normalization method with expressive backbone networks.
>
> 2. Figure 3 (caption): we discuss that full adaptivity can be obtained by incorporating RNF.
>
> 3. Section 3.1: We dedicate this subsection to discuss the design choices made in GRANOLA, and **emphasize the need for both expressivity and adaptivity**.
>
> 4. Proposition 4.1 and Lines 226-234: We discuss the necessity of expressiveness in addition to adaptivity.
>
> 5. Lines 929-932: We state that “GRANOLA benefits from (i) enhanced expressiveness, and (ii) graph adaptivity.”, and elaborate on this point within these lines.
>
> *In terms of experiments, we have extensively shown that:*
>
> 1. Adaptivity in itself improves the baseline results, as consistently shown by the variant called GRANOLA-NO-RNF.
>
> 2. Including RNF improves baselines results in many cases. (In particular the variants denoted by RNF-PE and RNF-NORM).
>
> 3. The combination of RNF and Adaptivity achieves the overall largest improvement compared to the baseline – this is our method GRANOLA.
>
> For convenience, we also provide several key results from the paper that support our claims discussed above, and also mention whether each method is adaptive/expressive. As can be seen, all directions (adaptive/expressive) help to improve the baseline, and having both properties achieve the best performance among these variants.
>
> | Method               | ZINC-12k $\downarrow$ | OGBG-MOLHIV $\uparrow$ | Adaptive | Expressive |
> |----------------------|-----------------------|------------------------|----------|------------|
> | GIN+BatchNorm        | 0.1630±0.004          | 75.58±1.40             | No       | No         |
> | GIN+BatchNorm+RNF-PE | 0.1621±0.014          | 75.98±1.63             | No       | Yes        |
> | GIN + RNF-NORM       | 0.1562±0.013          | 77.61±1.64             | No       | Yes        |
> | GIN + GRANOLA-NO-RNF | 0.1497±0.008          | 77.09±1.49             | Yes      | No         |
> | GIN + GRANOLA        | **0.1203±0.006**          | **78.98±1.17**             | **Yes**      | **Yes**        |
>
> We understand that the Reviewer feels that this point might have not been stressed enough in the paper, and in our revised version we will ensure to further highlight it. We thank you for the invaluable guidance.

---

> > ### Author Response · Authors · 2024-08-12
> > **Post-rebuttal follow-up (part 2)**
> >
> > **Regarding GRANOLA as a practical way to utilize RNF:** We thank the Reviewer for the comment, which we fully agree with and discuss in our original submission (page 6, “Relation to expressive GNNs”). We made this discussion even clearer in the revision.
> >
> > We believe this additional perspective on our contribution (i.e., "improving GNNs with random features" rather than "designing effective GNN normalization schemes") strengthens our work rather than diminishes it. Multiple viewpoints often emerge in scientific research and tend to enrich the overall discussion.

---

> > > ### Comment · Reviewer_Ao3Y · 2024-08-14
> > > **Recommendation**
> > >
> > > My overall perspective on this work remains the same (lacking some level of maturity + unclear insights - see the review and post-rebuttal comments above). However, I acknowledge that the authors have made a substantial effort to deepen their evaluation of their work during the rebuttal period. Therefore I decided to increase my rating to 6 and anticipate that the manuscript will be revised according to the discussion with the reviewers.

---

### Official Review · Reviewer_vndL · 2024-07-17

**Soundness:** 3
**Presentation:** 2
**Contribution:** 3
**Rating:** 5
**Confidence:** 3

**Summary:**

This paper proposes an adaptive normalization layer for GNNs. It first points out that the traditional normalization layer (BatchNorm, InstanceNorm)  are not specifically designed for graphs and thus may limit the expressive power of GNNs, and single normalization technique can not always be the best for all graphs. Therefore, it proposes the GRANOLA method which is a learnable normalization layer for GNNs, and justify its design with theories.

**Strengths:**

S1. the paper is well-structured and easy to follow.

S2. GRANOLA consistently bring performance improvement on all benchmark datasets.

S3. Baselines are carefully selected and organized.

**Weaknesses:**

W1. GRANOLA involves an additional GNN module to learn the adaptive normalization, making it more costly and less scalable than its counterparts. It would be better if the authors could provide some experimental results to show how much additional time we will need when applying GRANOLA.

W2. The proposed GRANOLA is only tested on GIN and GSN backbones. It would be better to check if it could be applied to more common backbone models such as GCN, GAT, etc.

W3. (This is a minor point). Code is not provided.

**Questions:**

Q1. The authors point out that the traditional normalization may not effectively capture the unique characteristics of graph-structured data. I am curious to see is there any empirical study or theoretical results to justify this point?

Q2. Figure 2 provides an example to illustrate that BatchNorm may make the GNN less powerful. Is it possible to have something similar to visualize the performance of BatchNorm, InstanceNorm, LayerNorm, and GRANOLA?

**Limitations:**

Please refer to weakness and questions.

---

> ### Author Rebuttal · Authors · 2024-08-07
>
> We thank the reviewer for their constructive feedback. We are glad to see the reviewer has appreciated the presentation of our work, finding the paper well-structured and easy to follow. We proceed by answering each question in the following.
>
> **W1:** GRANOLA involves an additional GNN module to learn the adaptive normalization, making it more costly and less scalable than its counterparts. It would be better if the authors could provide some experimental results to show how much additional time we will need when applying GRANOLA.
>
> **A:**  We agree with the reviewer that our GRANOLA involves some additional computations due to the GNN layers in the normalization mechanism. However, this addition allows the normalization to adapt to the input graph and the task at hand, yielding significant performance improvements as reflected in our experiments.
> Furthermore, in our paper, we report and discuss the runtimes in Appendix G, as well as the computational complexity of our method (which is linear in the number of nodes and edges, as standard MPNNs). Our results indicate that while GRANOLA requires additional computations (3x slower than Batchnorm), it is still a fraction of the cost of more complex methods (5.2x faster than a scalable provably-powerful GNN), while yielding favorable downstream performance. We believe that finding good tradeoffs between computational complexity of methods and their accuracy is important. In our revision, we will ensure to highlight this point better in the main text.
>
> **W2:** The proposed GRANOLA is only tested on GIN and GSN backbones. It would be better to check if it could be applied to more common backbone models such as GCN, GAT, etc.
>
> **A:** As stated in our paper on Lines 267-268, our motivation for experimenting with GIN is that it is maximally expressive among standard MPNNs (i.e., it is as expressive as 1-WL). Additionally, we demonstrated that our GRANOLA method can be beneficial for methods that exceed 1-WL expressiveness, such as GSN. This shows that GRANOLA is beneficial for different kinds of GNNs with varying levels of expressiveness.
> Nevertheless, we agree that experimenting with additional backbones is interesting and important. Therefore, we have now added results using GCN (Kipf et al., 2017), GAT (Veličković et al., 2018), and GPS (Rampášek et al., 2022) as backbones, combined with our GRANOLA method. Our results are provided in the table below and have been added to our paper.
> As can be seen from the table, GRANOLA **consistently improves** these various backbones. These results further underscore the versatility of GRANOLA, which can be potentially coupled with any GNN layer and improve its performance.
>
> | Method             | ZINC-12k  $\downarrow$  | OGBG-MOLHIV $\uparrow$ |
> |--------------------|-------------|-------------|
> | GCN                | 0.367±0.011 | 76.06±0.97  |
> | GCN+GRANOLA (Ours) | 0.233±0.005 | 77.54±1.10  |
> |--------------------|-------------|-------------|
> | GAT                | 0.384±0.007 | 76.0±0.80   |
> | GAT+GRANOLA (Ours) | 0.254±0.009 | 77.39±1.03  |
> |--------------------|-------------|-------------|
> | GPS                | 0.070±0.004 | 78.80±1.01  |
> | GPS+GRANOLA (Ours) | 0.062±0.006 | 79.21±1.26  |
>
>
> Kipf et al., 2017. Semi-Supervised Classification with Graph Convolutional Networks. ICLR 2017
>
> Veličković et al., 2018. Graph Attention Networks. ICLR 2018
>
> Rampášek et al., 2022. Recipe for a General, Powerful, Scalable Graph Transformer. NeurIPS 2022
>
> **W3:** (This is a minor point). Code is not provided.
>
> **A:** We agree that making the code public is important. To this end, our submission includes a statement that upon acceptance, we will release our code on GitHub. We confirm that this is indeed the case.
>
> **Q1:** The authors point out that the traditional normalization may not effectively capture the unique characteristics of graph-structured data. I am curious to see is there any empirical study or theoretical results to justify this point?
>
> **A1:** Traditional normalization techniques may result in a loss of expressiveness (the ability to compute certain functions) due to their lack of consideration of the underlying graph. For example, as we show in our paper (Figure 2), normalizing an MPNN with BatchNorm, together with the choice of the ReLU activation function (which is a very common choice used in various papers, as discussed in Lines 112-113), might lead to the loss of the ability to compute node degrees, which is an essential feature in graph learning.
>
> **Q2:** Figure 2 provides an example to illustrate that BatchNorm may make the GNN less powerful. Is it possible to have something similar to visualize the performance of BatchNorm, InstanceNorm, LayerNorm, and GRANOLA?
>
> **A2:** In Appendix C, we present additional motivating examples representing failure cases for other normalization methods. However, we understand the importance of visualization, and we will include additional figures similar to Figure 2. Thank you for the suggestion.
>
> ---
> We sincerely appreciate the reviewer’s constructive feedback. We have made efforts to address their questions thoroughly, and have conducted the suggested experiments. We kindly ask them to reconsider their rating if they find our responses satisfactory.

---

### Author Rebuttal · Authors · 2024-08-07

We would like to thank all reviewers for their valuable comments and for their efforts in providing actionable feedback to enhance the quality of the submission.

Overall, reviewers appreciated our novel adaptive normalization scheme for GNNs that maintains linear complexity and provides consistent improvements over all normalization schemes. We are glad to see that reviewers recognized the potential impact of the paper, found *“of general interest to the GNN community”* (**Ao3Y**), *“having good potential for adaptation in practice”* (**Ao3Y**),  being *“a practical solution for real-world applications”* (**tP2u**). The reviewers further highlighted the contribution of this work, deeming it *“significant”*  (**PwFk**), and our approach *“novel”* (**tP2u**, **PwFk**) .

Additionally, reviewers unanimously appreciated the experimental evaluation, finding it *“extensive”* (**tP2u**, **PwFk*), yielding *“consistent improvements”* (**vndL**, **Ao3Y**, **tP2u**) and recognizing its *“competitiveness with the state-of-the-art”* (**Ao3Y**). Additionally, we are glad to see reviewers **tP2u** and **PwFk** have both appreciated the theoretical analysis, finding it *“solid”* (**tP2u**, **PwFk**).

The paper presentation has also been particularly valued, with the paper described as  *“well-structured”* (**vndL**), *“well-written”* (**Ao3Y**),  and *“easy to follow”* (**vndL**, **Ao3Y**).

**New Experiments.** Several additional experiments were conducted following the reviewers’ comments:
1. An additional variant of our approach, where instead of using an MPNN + RNF as $\text{GNN}_\text{NORM}$, we employ a Subgraph GNN, showing that full-graph adaptivity can be obtained with any expressive architecture other than MPNN + RNF, and our choice was dictated by their linear complexity (**Ao3Y**);
2. A comparison of GRANOLA and standard normalizations using GAT as the GNN backbone, showing that GRANOLA is beneficial for different kinds of GNNs and consistently improves their performance (**vndL**);
3. A variant of our approach that uses the BatchNorm blueprint instead of the LayerNorm-node one, demonstrating that adaptivity is beneficial for different normalization blueprints (**Ao3Y**);
4. An additional analysis of the impact of the normalization term, achieved by removing the $\gamma_{b,n,c}^{(\ell)}$ term by setting it to zero, further highlighting the importance of normalizing (**tP2u**);
5. An evaluation of the performance of GRANOLA when coupled with graph transformers, demonstrating the versatility of GRANOLA, which can be coupled with any layer and improve its performance (**PwFk**).

If the reviewers find that we have adequately addressed their concerns, we would be grateful if they would consider reassessing their rating. We are also open to further discussion and welcome any additional input.

---

### Author Response · Authors · 2024-08-12
**Rebuttal Follow-up**

Dear Reviewers,

We sincerely appreciate your valuable feedback in your reviews, and thank Reviewer **PwFk** for raising their score following our rebuttal.

**We have made dedicated efforts to address all concerns raised by all reviewers in our rebuttal.
We look forward to receiving feedback from Reviewers vndL, Ao3Y, and tP2u, and hope that you will consider raising your scores as well based on our responses. We are also happy to discuss additional questions or suggestions you may have.**


Thank you for your time and consideration.

Best regards,

The Authors.

---

### Author Response · Authors · 2024-08-14

Dear Reviewers,

We would like to extend our gratitude for your insightful and detailed reviews, as well as the post-rebuttal feedback and discussions. In our final version, we will include all the discussions and results presented in our rebuttal and follow-up discussions.

With kindest regards,

The Authors.

---

### Decision · Program_Chairs · 2024-09-25

**Decision:**

Accept (poster)

**Comment:**

This paper proposes a new usage of RNF (random node features) for GNNs by not just concatenating the random features to node representations but using RNF + another MPNN to compute the affine parameters of LayerNorms in the main GNN, so that different nodes can be normalized differently. Empirically the new method shows improved performance compared to traditional normalization methods and the plain MPNN + RNF. Despite the uniform acceptance recommendations from reviewers, I would like to point out a few aspects to improve the paper: 1. RNF can only approximately maintain permutation equivariance with extensive training, and is thus less useful in practice. The authors could explain why RNF used for normalization can alleviate this issue. 2. The theoretical analysis is a bit trivial. I would expect a more in-depth analysis of the training dynamics with the proposed normalization.